# What makes math problems hard for reinforcement learning: a case study

Ali Shehper[1]    Anibal Medina-Mardones[2]    Lucas Fagan[3]    Bartłomiej Lewandowski[4]

Angus Gruen[5]    Yang Qiu[6]    Piotr Kucharski[4]    Zhenghan Wang[3]    Sergei Gukov[7]

[1]Department of Mathematics, California Institute of Technology
[2]Department of Mathematics, Western University
[3]Department of Mathematics, University of California, Santa Barbara
[4]Institute of Mathematics, University of Warsaw
[5]Polygon Zero    [6]Chern Institute of Mathematics and LPMC, Nankai University
[7]Richard N. Merkin Center for Pure and Applied Mathematics, California Institute of Technology
correspondence: mshehper@caltech.edu

## Abstract

Using a long-standing conjecture from combinatorial group theory, we explore, from multiple perspectives, the challenges of finding rare instances carrying disproportionately high rewards. Based on lessons learned in the context defined by the Andrews–Curtis conjecture, we analyze how reinforcement learning agents handle problems of varying hardness. We also address many mathematical questions as a part of our study. Notably, we demonstrate the length reducibility of all but two presentations in the Akbulut–Kirby series (1981), and resolve various potential counterexamples in the Miller–Schupp series (1991), including three infinite subfamilies.

## 1 Introduction

In recent years, Artificial Intelligence (AI) systems have demonstrated remarkable success in board games, video games, and other tasks requiring extensive planning and intelligence. Mathematics appears to be the next frontier, with increasing advancements in theorem-provers and theorem-proving assistants (Azerbayev et al., 2023; DeepMind, 2024; Yang et al., 2024; Lin et al., 2024). Mathematical problems can often be framed as search problems, where the goal is to find a path from a hypothesis to a conclusion through a sequence of basic logical steps.

For general mathematical problems, the action space is often vast—sometimes infinite—and solutions may require long sequences of steps, which may additionally be extremely rare in a large search space. As a result, mathematical problems, particularly those at the research level, present new challenges for AI systems and demand novel reinforcement learning algorithms.

At the same time, some mathematical problems—such as the Andrews–Curtis conjecture (Andrews & Curtis, 1965), which we study in this paper—offer distinct advantages for algorithm development compared to generic mathematical problems. First, the action space in this setting is finite and small, consisting of only a few basic moves, while the state space is infinite. Second, these problems come with a notion of *hardness*, with difficulty levels ranging from very easy to those requiring superexponential effort to resolve (Bridson, 2015; Lishak, 2017). Moreover, the distribution of states with respect to this measure is very non-uniform.

39th Conference on Neural Information Processing Systems (NeurIPS 2025).

These aspects provide unique opportunities for algorithmic development and may offer insights applicable to other complex problems, both in mathematics and beyond. We anticipate that future reinforcement learning algorithms will need to assess problem hardness during training, adapting dynamically to the most challenging instances they encounter.

In this paper, we initiate the study of problem hardness in this context, with the goal of developing novel algorithms. Our contributions include:

- **Framing the Andrews–Curtis trivialization problem as a reinforcement learning environment** characterized by long horizons, sparse rewards, and an intrinsic distribution of problem hardness.

- **Developing an understanding of the hardness distribution** for a large class of examples through intrinsic and path-based measures.

- **Analyzing how reinforcement learning agents handle problems of varying hardness** and using these observations to propose novel reinforcement learning algorithms that dynamically adapt to difficult challenges.

As a byproduct, we resolve various potential counterexamples to the conjecture that had remained open for more than 25 years (Miller & Schupp, 1999), including three infinite subfamilies, and discover new results for another family that has been open for over 44 years (Akbulut & Kirby, 1985). These results emerge from a collaboration between human mathematicians and computational systems, forming a feedback loop: mathematicians design computational tools; machines generate novel discoveries for specific instances; and mathematicians refine these discoveries into general results for infinite families.

These findings are of significant interest in mathematics and reinforce our belief that, in the coming years, reinforcement learning algorithms will play a major role in the discovery of new and important mathematical results.

## 2  Related Work

*Reinforcement Learning for long-horizon, sparse-reward environments.* Long-horizon, sparse-rewards problems have been a persistent challenge in reinforcement learning (RL), as they require agents to explore efficiently and learn from delayed and infrequent feedback. Atari games such as Montezuma's Revenge (Bellemare et al., 2013), which requires long-term planning to collect keys and open doors, and Pitfall! (Machado et al., 2018) have exemplified these difficulties. Similarly, robotic tasks such as dexterous manipulation (Rajeswaran et al., 2017) require agents to learn complex sequences of actions with little intermediate feedback.

Several approaches have been developed to tackle long-horizon, sparse-reward problems, including intrinsic motivation (Pathak et al., 2017), reward shaping (Ng et al., 1999), and hierarchical reinforcement learning (HRL) (Sutton et al., 1999). Intrinsic motivation techniques such as curiosity-driven exploration encourage agents to explore novel states even in the absence of external rewards. Reward shaping methods provide additional guidance to help bridge the sparse reward signal but require careful engineering to avoid unintended biases. HRL methods, including options frameworks (Sutton et al., 1999), hierarchical abstract machines (Parr & Russell, 1997), and feudal reinforcement learning (Dayan & Hinton, 1992; Vezhnevets et al., 2017) learn structured policies that decompose complex tasks into smaller sub-problems. More recently, memory-based approaches (Ecoffet et al., 2019; Badia et al., 2020) have been employed to improve long-term planning in these challenging environments.

*Andrews–Curtis Conjecture.* Previous studies of the Andrews–Curtis conjecture have used various search algorithms, including genetic algorithms (Miasnikov, 2003a), breadth-first search (Havas & Ramsay, 2003), and other more sophisticated algorithms (Bowman & McCaul, 2006; Panteleev & Ushakov, 2019; Krawiec & Swan, 2016).

# 3 Background: The AC Conjecture

## 3.1 Mathematical statement

The Andrews–Curtis conjecture is a long-standing conjecture in combinatorial group theory (Andrews & Curtis, 1965) concerning *balanced presentations* of the trivial group.

A group presentation $\pi = \langle x_1, \ldots, x_m \mid r_1, \ldots, r_p \rangle$ consists of a list of generators (the $x_i$'s) and a list of relators (the $r_j$'s), where each relator is a word over the alphabet $x_1^{\pm 1} \cdots x_m^{\pm 1}$. The *length* of $\pi$, denoted $\ell(\pi)$, is the sum of the word lengths of all the relators, and $\pi$ is said to be balanced if $m = p$. As we only consider balanced presentations here, we will usually omit the adjective.

The Andrews–Curtis (AC) conjecture states that any balanced presentation of the trivial group is *AC-equivalent* to the trivial presentation: $\langle x_1, \ldots, x_m \mid x_1, \ldots, x_m \rangle$. That is, any such presentation can be transformed into the trivial one through a sequence of operations known as *AC-moves*:

1. Substitute some $r_i$ by $r_i r_j$ for $i \neq j$.
2. Replace some $r_i$ by $r_i^{-1}$.
3. Change some $r_i$ to $x_j^{\pm 1} r_i x_j^{\mp 1}$.

If such a sequence of moves exists, we say that the starting presentation can be *trivialized*. In this work, we also consider a modified set of moves, which we call *AC'-moves*.

1. Replace some $r_i$ by $r_i r_j^{\pm 1}$ for $i \neq j$.
2. Change some $r_i$ to $x_j^{\pm 1} r_i x_j^{\mp 1}$.

The system of balanced presentations with AC-moves is equivalent to the system of balanced presentations with AC'-moves, as we show in Appendix F. The Andrews–Curtis conjecture may be studied for any $m$. Here, we focus on the case $m = 2$ and denote the generators as $x, y$ instead of $x_1, x_2$.

## 3.2 Potential counterexamples

Any presentation of length less than 13 is known to be consistent with the conjecture (Miasnikov, 2003b; Havas & Ramsay, 2003). The shortest potential counterexample is the element AK(3) in the following infinite family of potential counterexamples known as the *Akbulut–Kirby series* (Akbulut & Kirby, 1985):

$$\mathrm{AK}(n) = \langle x, y \mid x^n = y^{n+1},\ xyx = yxy \rangle.$$

Another infinite family containing many potential counterexamples of interest is known as the *Miller–Schupp series* (Miller & Schupp, 1999):

$$\mathrm{MS}(n, w) = \langle x, y \mid x^{-1} y^n x = y^{n+1},\ x = w \rangle.$$

Here, $n \geq 1$ and $w$ is a word in $x$ and $y$ with zero exponent sum on $x$. The two families are known to be related: AK($n$) is AC-equivalent to MS($n$, $y^{-1} x^{-1} y x y$) for all $n$ (Myasnikov et al., 2002).

## 3.3 AC graph and path-based hardness measures

The *AC graph* consists of nodes and edges labeled respectively by presentations and AC moves. In this language, the conjecture states that, for any node, there is a path connecting it to the node representing the trivial presentation. Such a path is called a *trivialization*.

To measure how hard a presentation is to trivialize, we choose a cost function for paths and minimize it over all trivializations of the given presentation. A natural cost function is the number of edges in a path, called the *path length*. Another option is $\ell$-*increase*, which is defined for a path $\gamma = (\pi_0, \ldots, \pi_N)$ by $\max_{i \in \{0, \ldots, N\}} \ell(\pi_i) - \ell(\pi_0)$.

# 4 Methods

## 4.1 Benchmark dataset

We evaluate our methods by counting the number of presentations solved from the benchmark dataset $\mathcal{D}$. We consider Miller–Schupp presentations MS($n, w$) with $n \leq 7$ and Length($w$) $\leq 7$ up to

trivial identifications (see Appendix D). The resulting dataset contains 1190 presentations, with a maximal presentation length of 25. The methodology for constructing the dataset and the full list of presentations are provided in Appendix B.

## 4.2 Classical search algorithms

We use two classical search algorithms—breadth-first search (BFS) and greedy search (GS)—and we use AC$'$-moves to search for AC trivializations. [1] [2] In GS, the state with the smallest presentation length $\ell$ is chosen, with ties being broken by the path length $l$ from the current state to the initial state.

Since the AC graph is infinite, an unconstrained search is impractical. We restrict the search to presentations whose relators have word lengths at most 512. Despite this restriction, the search space remains vast. To prevent memory outages, we limit each algorithm to visiting a maximum of 1 million nodes.

## 4.3 Reinforcement learning

The AC trivialization problem can be framed as a sequential decision-making task within the Markov Decision Process (MDP) framework, enabling the application of reinforcement learning algorithms to solve it.

An MDP is characterized by five key components: the state space, action space, transition probability function, initial state distribution, and reward function. We now outline the specific choices made in formalizing the Andrews–Curtis problem as an MDP and empirically evaluating the performance of reinforcement learning agents on this problem.

*State space*. The space of balanced presentations of the trivial group. As this state space is infinite, we impose an upper limit of 512 on the word length of each relator to make it finite.

*Action space*. We consider two choices for the action space: either the set of AC or AC$'$ moves.

*Transition probabilities*. The transition probability function is deterministic. The trivial presentation, which has the shortest possible length $\ell = 2$, is a *terminal state* of the MDP. Reaching a terminal state successfully solves the initial presentation of a trajectory and ends the episode.

*Initial state distribution*. When attempting to solve presentations from the benchmark dataset $\mathcal{D}$, we randomly sample from $\mathcal{D}$ following a uniform distribution.

*Reward function*. The AC conjecture does not come with a fixed reward function. One choice is to assign $+1$ to a transition into a terminal state, indicating that a presentation has been successfully solved, and 0 to every other transition. This reward function, denoted $R_1$, is extremely sparse. We experiment with providing intermediate rewards that penalize transitions into long presentations,

$$R_2(s_t, a_t, s_{t+1}) = \begin{cases} -\ell(s_{t+1}) & \text{if } \ell(s_{t+1}) > 2, \\ 1024 \cdot T_{\max} & \text{otherwise.} \end{cases}$$

The reward for the terminal transition $1024 \cdot T_{\max}$ is chosen to counteract the accumulated penalties from long paths. Specifically, when the discount factor $\gamma = 1$, the lowest bound on the total return of any given trajectory is $-1024 \cdot T_{\max}$ as each of the two relators is allowed a maximum word length of

---

[1] We tested a few more search algorithms including $A^*$ search and MCTS search but the results were underwhelming.

$A^*$ search is a half-way option between breadth first and greedy search, where the priority queue is sorted by the sum of the path length $p$ and presentation length $l$. We tested a family of search algorithms that sorted the priority queue using $rp+l$ for a variety of choices of $r = 0, 1 \cdots, 10$ and measured how many MS presentations each search algorithm solved. We found that decreasing $r$ improved the number of presentations solved with the maximum being at $r = 0$ which corresponds to greedy search.

As for MCTS, we observed that it spends a meaningful amount of time exploring different paths which do not lead to a trivialization solution. In contrast, the length heuristic function of greedy search helps find AC trivializations much faster, and with a much smaller usage of system memory.

[2] AC$'$-moves provide an advantage over AC-moves in GS when nodes are ordered using presentation length $\ell$. Each AC$'$-move affects $\ell$, providing a signal to the search process. In contrast, the inversion move (AC2) leaves $\ell$ invariant.

512. The large terminal reward precisely cancels out this worst-case sum, thereby ensuring that a successful trajectory achieves a positive total return.

We also consider a third reward function $R_3$ that clips the negative rewards at $-10$ and positive rewards at $1000$.

*Maximum horizon length.* Solving a Markov Decision Process requires specifying a maximum horizon length $T_{\max}$ and a discount factor $\gamma$. The horizon length $T_{\max}$ is a key variable in our problem as presentations requiring AC paths of lengths longer than $T_{\max}$ would necessarily remain unsolved by any agent. Simultaneously, there exist presentations for which the path length is a superexponential function of the presentation length (Bridson, 2015; Lishak, 2017), ensuring that Andrews–Curtis MDP is a long-horizon problem.[3]

The efficacy of our algorithms depends on our choice of $T_{\max}$. Hence, we experiment with different schedules for it.

1. Constant Schedule: We keep $T_{\max}$ at a constant value throughout the training of a reinforcement learning agent, using $T_{\max} = 200$ and $T_{\max} = 400$.

2. Piecewise Constant Schedule: We vary $T_{\max}$ during training as follows. For an agent trained for 100M environment interactions, we start $T_{\max}$ at 200, increase to 400 after 10M interactions, 800 after 25M, and 1200 after 50M. We anticipate that the gradual increase in $T_{\max}$ in the latter case will allow the agent to first master solving simpler presentations before progressively tackling more complex ones as its abilities develop.

On the surface, these values seem much smaller than horizon lengths for various video game environments that reinforcement learning algorithms are known to excel at. However, a difference between games and math environments is that there is no way to fail early during an episode and hence for an agent to learn what steps *not* to take.

*Algorithms.* We use Deep Q-Learning (DQN) Mnih et al. (2015), Synchronous Actor-Critic (A2C) (Mnih, 2016; Schulman et al., 2017), Proximal Policy Optimization (PPO) (Schulman et al., 2017), and AlphaZero Silver et al. (2017, 2018) to solve the Andrews–Curtis MDPs.

*State encoding.* We encode each relator of a presentation as an array of length $512$. The generators and their inverses — $x$, $y$, $x^{-1}$, and $y^{-1}$, are represented by integers $1$, $2$, $-1$, and $-2$ respectively. If the length of the relator is less than $512$, the array is padded to the right with zeros. A balanced presentation of two relators is represented by an array of length $1024$. AC and AC′ moves are implemented as functions on such arrays. If a move results in a presentation with a relator of length more than $512$, it is taken to act trivially instead.

*Training setup.* In our preliminary experiments of A2C, PPO and AlphaZero (discussed in Section 5.1), we use feedforward neural networks with two hidden layers of $512$ neurons each for both the actor and critic networks. The actor and critic networks do not share any parameters. In DQN experiments, we use feed-forward neural networks with two hidden layers of $842$ neurons so that the number of parameters is roughly the same. A2C, PPO and DQN agents are trained for 100M environment interactions. AlphaZero agents are trained for approximately 5.7M environment steps with 32 simulations per environment step, taking approximately the same amount of wall-clock time as PPO training runs. We use reward-normalization for on-policy algorithms (i.e. A2C and PPO).

In later experiments (see Section 5.2), we use residual networks consisting of six residual layers to train PPO agents for 1B environment interactions. Additional details on the network architectures and experimental setup are provided in Appendix C. For each training configuration, we train three agents with different random seeds and report the mean, minimum, and maximum performance.

---

[3]Roughly speaking, for these presentations, the number of AC-moves required to trivialize the presentation is at least $\Delta(\lfloor \log_2 \ell \rfloor)$ where $\Delta \colon \mathbb{N} \to \mathbb{N}$ is defined recursively as $\Delta(j) = 2^{\Delta(j-1)}$ for $j \geq 1$ and $\Delta(0) = 2$. For a considerably small value $\ell = 13$, $\Delta(\lfloor \log_2(13) \rfloor) = 65536$.

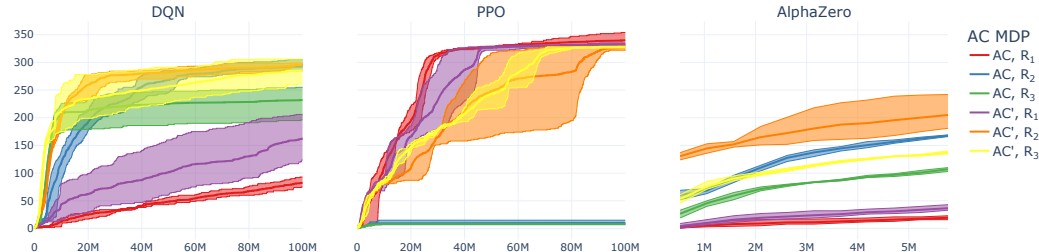

Figure 1: Comparison of RL agents across AC MDPs. Number of presentations solved from the benchmark dataset $\mathcal{D}$ (Y-axis) versus number of environment steps (X-axis).

## 5 Results

### 5.1 RL: Comparing algorithms, rewards, and actions

For the two choices of action spaces and three choices of reward functions, we plot the performance of DQN, PPO and AlphaZero agents in Figure 1. The best choice of action space and reward function depends on the choice of algorithm.

For either choice of action space, DQN struggles with sparse rewards ($R_1$). Introducing intermediate rewards ($R_2$) helps the performance, while clipping rewards ($R_3$) hurts the performance compared to dense and un-clipped rewards. This indicates that the agent benefits from distinguishing between presentations with total length greater than 10. Overall, the best DQN agent solves 303 presentations.

PPO agents do not benefit from intermediate rewards in the same way as DQN. Sparse and dense rewards lead to comparable levels of performance when training with AC$'$ action space. Clipping rewards ($R_3$) reduces variance in training curves across different seeds, but it does not affect the mean performance. Over the entire training period, four out of six PPO agents perform approximately at the same level with the two exceptions being agents trained with AC moves and reward functions $R_2$ and $R_3$. These reward functions depend on the length of the presentation, which the inversion move (AC2) leaves invariant. In these cases, agents settle early into a suboptimal policy, making repeated use of the AC2 move, and hence not fully exploring the state space. We also note that agents trained with AC moves and $R_1$ reward function are most sample-efficient. The best PPO agent solves 353 presentations.

As with DQN, AlphaZero performs poorly with sparse reward function ($R_1$). The $y$-intercept in Figure 1 (right) specifies the number of presentations solved by MCTS with randomly initialized policy and value networks. When training with dense reward function $R_2$, the search process is able to identify actions that give shorter presentations, resulting in larger values of $y$-intercept. Overall, the best AlphaZero agent solves 242 presentations.

Finally, we remark that performances of A2C agents follows the same pattern across AC MDPs as the PPO agents. This is explained by a previous observation that A2C is a special case of PPO Huang et al. (2022c). We present a comparison of the best-performing A2C and PPO agents, both trained with sparse reward function $R_1$ and AC action space, in Appendix C.

Our preliminary results indicate that PPO outperforms DQN, A2C, and AlphaZero on the AC problem. Coupled with the AC environment's low computational cost, which makes it well-suited to large-scale on-policy training, we therefore concentrate on PPO in the remainder of this paper.

### 5.2 BFS vs. GS vs. PPO

We compare the performance of BFS, GS, and PPO on the task of solving presentations in the benchmark dataset $\mathcal{D}$. Out of the 1190 presentations in the dataset, BFS and GS solve 278 and 533 presentations respectively.

For PPO with constant horizon length schedule, the best agent with $T_{\max} = 200$ solved $457$ presentations, and the best agent with $T_{\max} = 400$ solved a subset of these presentations of size $402$. With a smaller horizon length, the agent spends less time exploring unpromising directions.

The PPO agent with $T_{\max} = 200$ solves five presentations that GS fails to solve. These cases require a substantial increase in $\ell$ along the trivialization path—something GS cannot accommodate due to its greedy behavior with respect to $\ell$. In contrast, PPO learns an effective exploration strategy over the AC state space, enabling it to discover these non-greedy trivialization paths.

When varying horizon length with the piecewise-constant schedule described above, the best PPO agent solved two new presentations from $\mathcal{D}$ that all other algorithms (including PPO with constant $T_{\max}$) are unable to solve. We provide a complete list of all presentations solved by all algorithms in Appendix B. The AC trivialization paths for all presentations solved by our algorithms are included in the supplementary material, which also includes the code to verify the correctness of these paths and to conduct the experiments discussed here.

### 5.3 Theoretical results

Our agents made two significant mathematical contributions: first, they discovered paths that resolved long-standing potential counterexamples to the Andrews-Curtis conjecture, and second, they revealed patterns in these paths that led us to formulate and subsequently prove new conjectures about infinite families of presentations. We state these results here, detailing their proofs in Appendix A.

*Theorem* A. The following infinite subfamilies of Miller–Schupp presentations are AC-trivial:

1. $\text{MS}(1, w)$ for all $w$.

2. $\text{MS}(n, y^{-1}xyx^{-1})$ for all $n$.

3. $\text{MS}(2, y^{-k}x^{-1}yxy)$ for all $k$.

*Theorem* B. For every $n \geq 2$, $\text{AK}(n)$ is AC-equivalent to the presentation

$$\langle x, y \mid x^{-1}yx = xyx^{-1}y \, , \, xy^{n-1}x = yxy \rangle,$$

of length $n + 11$. This gives a reduction in length of $AK(n)$ for all $n \geq 5$.

## 6 Hardness distribution

In this section, we study the distribution of hardness for presentations in the dataset $\mathcal{D}$ using various measures.

### 6.1 Intrinsic measures: $\ell$ and $n$

The goal state in the AC trivialization problem, i.e. the trivial presentation, has the shortest possible length. Hence, a natural intrinsic hardness measure for a presentation is its length $\ell$.

Figure 2a provides evidence that longer presentations are harder to trivialize. The relationship, however, is not predictive: $\text{MS}(2, yx^2yx^{-2})$ with $\ell = 14$ is not trivialized by any of our algorithms, but $\text{MS}(6, y^{-1}x^{-1}y^{-2}xy^2)$ with $\ell = 23$ is trivialized by both GS and PPO.

In the case of infinite families $\text{MS}(n, w)$ and $\text{AK}(n)$, another natural hardness measure is the integer $n$. The dataset $\mathcal{D}$ contains 170 presentations for each $n \in \{1, 2, \cdots, 7\}$. Figure 2b shows that as $n$ increases, the performance of our algorithms consistently declines.

### 6.2 Path-based measures

We use two path-based measures—path length and $\ell$-increase—to study hardness distributions of presentations solved by GS and PPO. In the case of PPO, we study the agent with $T_{\max} = 200$ from Section 5.2, which solved 457 presentations.

#### 6.2.1 Path length

We plot the distribution of path lengths for AC trivializations discovered by GS and PPO in Figure 3a and Figure 3b, respectively. In both cases, the distributions appear nearly continuous. Presentations

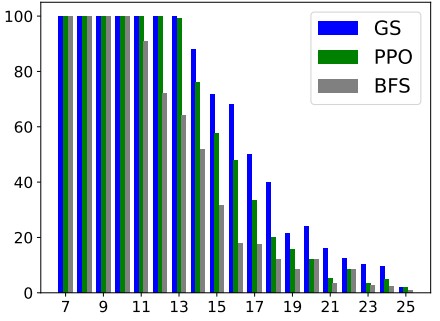
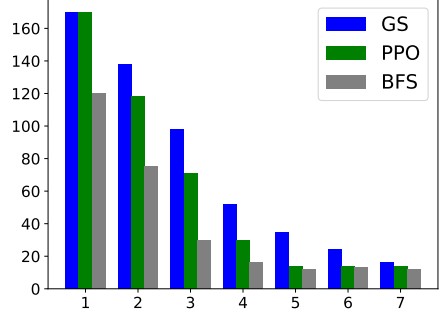

(a) Percentage of presentations solved versus length $\ell(\mathrm{MS}(n, w))$.

(b) Number of presentations solved versus $n$ of $\mathrm{MS}(n, w)$.

Figure 2: Distributions of the number of presentations solved by BFS, GS, and PPO with constant horizon length schedule as functions of lengths of the presentations and $n$.

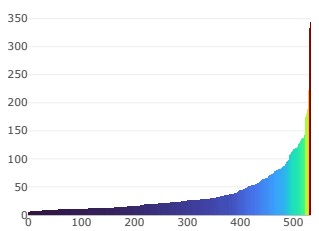
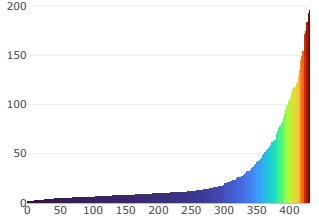
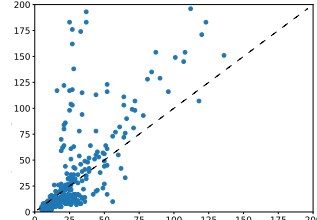

(a) Bar plot of path lengths discovered by GS. Path length (Y-axis) against problem instances (X-axis).

(b) Bar plot of path lengths discovered by PPO. Path length (Y-axis) against problem instances (X-axis).

(c) Path lengths discovered by PPO (Y-axis) against GS (X-axis) for presentations solved by both.

Figure 3: Distributions of path lengths discovered by GS and PPO, and their comparison.

requiring longer paths—indicative of higher complexity—are highlighted in red, while a long tail of easier presentations is shown in other colors.

A comparison of path lengths for presentations solved by both PPO and GS is shown in Figure 3c. For many easy presentations, PPO learns shorter paths compared to GS. However, as presentations become more difficult, the paths discovered by PPO tend to be longer than those found by GS— reflecting that while our agents are trained to find trivialization paths, they are not trained to find the shortest paths possible.

### 6.2.2 $\ell$-increase

We plot the distribution of $\ell$-increase for AC trivializations discovered by GS and PPO in Figure 4a and Figure 4b, respectively.

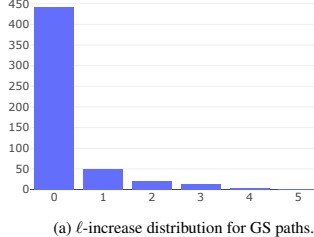
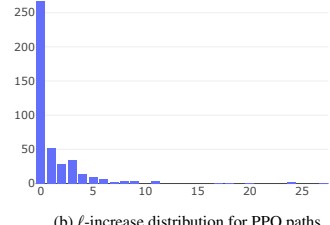

(a) $\ell$-increase distribution for GS paths.

(b) $\ell$-increase distribution for PPO paths.

Figure 4: Distributions of $\ell$-increase values in paths discovered by GS and PPO. Number of presentations (Y-axis) against $\ell$-increase (X-axis).

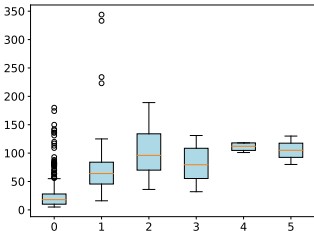

Figure 5: Path lengths of trivializations found by GS (Y-axis) as a function of $\ell$-increase (X-axis) for Miller-Schupp presentations $MS(n, w)$.

GS prioritizes states based on the pair $(\ell, l)$ (cf. Section 4.2), ensuring that it discovers paths with minimal $\ell$-increase values. As shown in Figure 4a, GS paths exhibit an $\ell$-increase of zero in most cases, with a maximum value of only 5. In contrast, PPO paths span a much wider range of $\ell$-increase values—ranging from 0 to 27, as depicted in Figure 4b. Thus, we observe that path-based hardness measures may rely heavily on the choice of the algorithm.

### 6.2.3 Relationship between path length and $\ell$-increase.

For presentations solved by GS we plot the solutions' path length as as a function of $\ell$-increase (Figure 5). Path lengths increase proportionally with $\ell$-increase, indicating the presence of a correlation between the two hardness measures. While this trend is visible in the case of GS due to its greedy selection with respect to $\ell$, there does not exist a similar pattern for paths discovered by PPO.

## 7 The cure: new algorithms

In this section, we summarize our insights and present new ideas for RL algorithm development based on the lessons learned in the case study at hand. Specifically, we argue that the following two elements can be useful in Andrews–Curtis MDPs and in other long-horizon problems with sparse rewards and hardness distribution of problem instances:

- Supermoves,
- Adaptive action spaces.

The former refers to actions that are compositions of basic moves. These appear to be a necessary, unavoidable tool to overcome superexponential lengths of sought-after solutions. Indeed, if the solution length is superexponential in the presentation length, $\ell$, the only way to turn it into a polynomial sequence of steps is to allow steps to grow in size, aggregating certain sequences of actions into new actions. We call these new actions of growing size "supermoves".

Supermoves are closely related to the ideas of compound actions and options often studied in the subject of hierarchical reinforcement learning. Our proposal, however, is that in long-horizon, sparse-reward problems where problem instances follow a distribution with respect to certain hardness measures, we may use the understanding of hardness itself to select new actions (supermoves) dynamically during training.

A practical implementation can be as follows. Consider one of the standard RL algorithms such as PPO with a fixed action space, such that some of the actions are initially masked. In the process of training, we gradually unmask these actions and replace them with supermoves at $10^N$ epochs, with $N = 1, 2, 3, \ldots$. During the intervals between the changes, we effectively train an RL model with a fixed action space. For the training history at previous stages to be useful at stage $N$, the changes to the action space at each stage must not be too large. The fraction of the action space that undergoes a change is a hyperparameter of the proposed algorithm, and it may be suitable to keep it in the range $5 - 15\%$. At each stage, labeled by $N$, one can take $n$ hardest presentations solved by the RL agent with the action space $A_N$ and add their successful solutions or some subsequences of these successful solutions as supermoves to the action space. This process is illustrated by a diagram in Figure 6a.

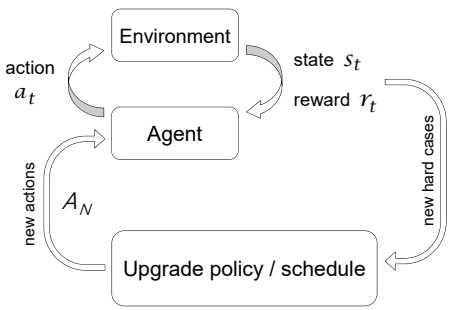

(a) A schematic representation of a model with an adaptive action space.

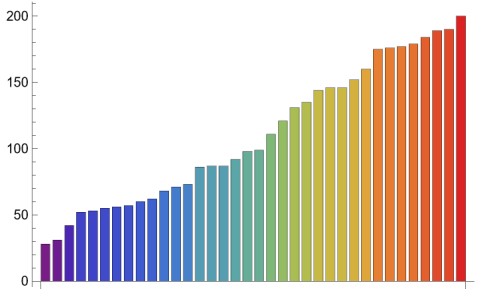

(b) Path length distributions of newly solved instances; hard instances are shown in red.

Figure 6: Visualizations of key concepts for supermoves and adaptive action spaces.

The implementation relies heavily on the notion of hardness with respect to a very specific underlying RL model at hand. Thus, if the standard part of the RL cycle in Figure 6a is based on a PPO algorithm with a particular choice of hyperparameters, then we need hardness with respect to this particular model.

As discussed in the previous sections, a good proxy for hardness can be the length of the solution path measured in basic elementary steps. In fact, we observe in preliminary experiments that at a given stage $N$, a typical distribution of this measure among solved examples often looks like Figure 6b, with hard instances shown in red. (See Figure 3b for the distribution of path lengths in an actual PPO agent, and Figure 10 in Appendix E for evidence that this behavior persists across scale, even as the total number of environment interactions vary.) For this hardness measure, we propose choosing sub-sequences of AC paths for paths shown in red (hard), expecting that in the space of all sequences of actions of a fixed length, these subsequences are both harder and more useful for the agent to learn.

**Empirical evidence.** While we leave a complete empirical investigation of these ideas to future work, we will describe some experiments that exhibit the utility of dynamic action spaces. Our setup is as follows: we enhance the action space of 12 AC$'$ moves to include all conjugations of the form $r_i \rightarrow w r_i w^{-1}$ for $i = 1, 2$, where $w$ is any word of length at most $L$. [4] All actions are initially masked except for the original 12 AC$'$ moves.

As the agent is trained, we look for sub-sequences of actions in its discovered solutions which are of the type $r_i \rightarrow w r_i w^{-1}$ and unmask actions corresponding to these supermoves. We repeat this process after every 100 epochs. Further details and hyperparameters of the training setup are described in Appendix C.

The agent solves 495 presentations including 13 new presentations that had eluded greedy search and PPO agents of Section 5.2. We highlight these presentations in the table provided in Appendix B and provide a more detailed analysis of the discovered paths in Appendix D.

## 8   Limitations

This study has two primary limitations. First, the training dataset was limited to 1190 presentations of the Miller–Schupp series. We expect that a larger and more diverse set of balanced presentations would yield more capable agents. Second, our work is focused on the Andrews-Curtis conjecture. While we believe the lessons learned are broadly applicable, the generalizability of our findings to other mathematical problems requires further empirical investigation.

## Acknowledgments and Disclosure of Funding

We would like to thank Danil Akhtiamov, Anna Beliakova, Jessica Craven, Michael Douglas, Konstantin Korovin, Miriam Lipniacka, Alexei Lisitsa, Maksymilian Manko, Ciprian Manolescu,

---

[4]$L$ is a new hyperparameter, which we set to 6. For general $L$, there are $2 \times 3^L - 2$ choices of $w$. The new action space thus has $4 \times 3^L - 4 + 4 = 4 \times 3^L$ actions. For $L = 6$, this amounts to 2916.

Fabian Ruehle, Michele Tarquini, Josef Urban, Richard Wedeen, and Tony Yue Yu for insightful discussions and comments. We especially want to thank Anna Beliakova for igniting our interest in the Andrews–Curtis conjecture as a framework for exploring problems with long and rare sequences of moves that an RL agent must discover.

A.M.'s work is supported by NSERC grants RES000678 and R7444A03. The work of P.K. and B.L. has been supported by the Polish National Science Centre through Sonata grant (2022/47/D/ST2/02058). Y.Q. is supported by National Key R&D Program of China (2024YFA1013202) and Nankai Zhide Foundation.

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

# NeurIPS Paper Checklist

The checklist is designed to encourage best practices for responsible machine learning research, addressing issues of reproducibility, transparency, research ethics, and societal impact. Do not remove the checklist: **The papers not including the checklist will be desk rejected.** The checklist should follow the references and follow the (optional) supplemental material. The checklist does NOT count towards the page limit.

Please read the checklist guidelines carefully for information on how to answer these questions. For each question in the checklist:

- You should answer [Yes] , [No] , or [NA] .
- [NA] means either that the question is Not Applicable for that particular paper or the relevant information is Not Available.
- Please provide a short (1–2 sentence) justification right after your answer (even for NA).

**The checklist answers are an integral part of your paper submission.** They are visible to the reviewers, area chairs, senior area chairs, and ethics reviewers. You will be asked to also include it (after eventual revisions) with the final version of your paper, and its final version will be published with the paper.

The reviewers of your paper will be asked to use the checklist as one of the factors in their evaluation. While "[Yes] " is generally preferable to "[No] ", it is perfectly acceptable to answer "[No] " provided a proper justification is given (e.g., "error bars are not reported because it would be too computationally expensive" or "we were unable to find the license for the dataset we used"). In general, answering "[No] " or "[NA] " is not grounds for rejection. While the questions are phrased in a binary way, we acknowledge that the true answer is often more nuanced, so please just use your best judgment and write a justification to elaborate. All supporting evidence can appear either in the main paper or the supplemental material, provided in appendix. If you answer [Yes] to a question, in the justification please point to the section(s) where related material for the question can be found.

IMPORTANT, please:

- **Delete this instruction block, but keep the section heading "NeurIPS Paper Checklist",**
- **Keep the checklist subsection headings, questions/answers and guidelines below.**
- **Do not modify the questions and only use the provided macros for your answers**.


# A  Proofs of Mathematical Results

In this appendix, we provide proofs of all mathematical results discovered in this work. We re-state our main results as two theorems.

*Theorem* C.  The following infinite subfamilies of Miller–Schupp presentations are AC-trivial:

1. $\text{MS}(1, w)$ for all $w$.

2. $\text{MS}(n, y^{-1}xyx^{-1})$ for all $n$.

3. $\text{MS}(2, y^{-k}x^{-1}yxy)$ for all $k$.

*Theorem* D.  For every $n \geq 2$, $\text{AK}(n)$ is AC-equivalent to the presentation

$$\langle x, y \mid x^{-1}yx = xyx^{-1}y \ , \ xy^{n-1}x = yxy \rangle,$$

of length $n + 11$. This gives a reduction in length of $AK(n)$ for all $n \geq 5$.

The three parts of Theorem C are proven respectively as Theorems A.2 and A.3 and Corollary A.9. Theorem D is proven as Theorem A.4.

## A.1  The Substitution Move

Our proofs rely on the following *substitution* move which comes from Burns & Macedonska (1993).

**Definition A.1** (Substitution).  Let $\langle x_1, \ldots, x_m \mid r_1, \ldots, r_{i-1}, w^{-1}w', r_{i+1}, \ldots r_m \rangle$ be a balanced presentation of the trivial group with some words $w$, $w'$ in generators. By a sequence of AC-moves, we may replace any occurrence of $w$ in a relator $r_j$ (where $j \neq i$) with $w'$.

Substitution move is a sequence of simple AC-moves. Consider by means of an example substituting $w'$ for $w$ in the relator $w_1ww_2$: first, conjugate to get $w_2w_1w$; then multiply by $w^{-1}w'$ to write $w_2w_1w'$; and finally, conjugate again to get the required result $w_1w'w_2$. For a relator of the form $w_1ww_2 \ldots w_{l-1}ww_l$, we may similarly substitute any number of occurences of $w$ with $w'$.

## A.2  AC-triviality of two MS families

**Theorem A.2.**  *$MS(1, w)$ is AC-trivial for all $w$.*

*Proof.*  We have

$$\text{MS}(1, w) = \langle x, y \mid x^{-1}yx = y^2, x = w \rangle$$

where $w$ has exponent sum 0 in $x$. We can re-write the first relator in the following four ways:

1. $yx = xy^2$

2. $x^{-1}y = y^2x^{-1}$

3. $x^{-1}y^{-1} = y^{-2}x^{-1}$

4. $y^{-1}x = xy^{-2}$

We can substitute these equations in the second relator to move around all occurrences of $x$ and $x^{-1}$: relations (i) and (iv) move $x$ to the left, relations (ii) and (iii) move $x^{-1}$ to the right. We continue until the second relator becomes $x^{a-1}y^bx^{-a}$ for some $a, b \in \mathbb{Z}$. Through conjugation, this simplifies to $x = y^b$, which when substituted into the first relator leads to a trivialization of the presentation.  $\square$

The following result follows from this theorem.

**Theorem A.3.**  *$MS(n, w_\star)$ for $w_\star = y^{-1}xyx^{-1}$ is AC-trivial for all $n$.*

*Proof.* In the presentation,

$$\text{MS}(n, w_\star) = \langle x, y \mid x^{-1}y^n x = y^{n+1}, x = y^{-1}xyx^{-1}\rangle$$

we can rewrite the second relation as $y^{-1}xy = x^2$, and apply the automorphism $x \leftrightarrow y$ to get

$$\text{MS}(n, w_\star) = \langle x, y \mid x^{-1}yx = y^2, y^{-1}x^n y = x^{n+1}\rangle.$$

This presentation is the same as $\text{MS}(1, y^{-1}x^n yx^{-n})$, which is AC-trivial by Theorem A.2. $\qquad\square$

AC-trivializations of $\text{MS}(n, w_\star)$ for $n = 3, 4, 5, 6, 7, 8$ were recently obtained using automated theorem proving in Lisitsa (2024). Here, we have obtained AC-trivializations of this family for all $n$.

## A.3 Length reduction for AK($n$)

**Theorem A.4.** *For every $n \geq 2$, AK($n$) is AC-equivalent to the presentation*

$$\langle x, y \mid x^{-1}yx = xyx^{-1}y \,,\; xy^{n-1}x = yxy\rangle,$$

*of length $n + 11$. This gives reduction in length of $AK(n)$ for all $n \geq 5$.*

To prove this theorem, we will use the following result due to Myasnikov et al. (2002), and prove two additional theorems (Theorem A.6 and Theorem A.7). From these results, Theorem A.4 follows immediately.

**Proposition A.5** (Myasnikov, Myasnikov, and Shpilrain, (Myasnikov et al., 2002))**.** *For all $n \geq 2$, $AK(n)$ is AC-equivalent to $MS(n, w_1)$ where $w_1 = y^{-1}x^{-1}yxy$.*

**Theorem A.6.** *For each fixed $n > 0$, the 1-parameter family of presentations $MS(n, w_k)$ are all AC-equivalent. Here, $w_k = y^{-k}x^{-1}yxy$ is parameterized by $k \in \mathbb{Z}$.*

*Proof.* Starting with the presentation,

$$\text{MS}(n, w_k) = \langle x, y \mid x^{-1}y^n x = y^{n+1}, x = y^{-k}x^{-1}yxy\rangle,$$

we will show that for any fixed $n$ and $k$, $\text{MS}(n, w_k)$ is AC-equivalent to $\text{MS}(n, w_{k+1})$.

First, note that the first relation in $\text{MS}(n, w_k)$ can be rearranged in the following three ways:

1. Multiplying by $y^{k-n}x$ from the left and by $y^{-1}$ from the right gives

$$y^k xy^{-1} = y^{k-n}xy^n.$$

2. Multiplying by $y^{-1}$ from the left and by $x^{-1}y^{-1}$ from the right, and inverting the relation gives

$$y^{-(n-1)}xy = yxy^{-n}.$$

3. Multiplying by $y^{-1}x$ from the left and by $y^{k-n}$ from the right, and inverting the relation gives

$$y^{n-k}x^{-1}y^{-(n-1)} = y^{-(k+1)}x^{-1}y.$$

Now, we can rearrange the second relation to get $y^k xy^{-1} = x^{-1}yx$. Substituting (i) gives $y^{k-n}xy^n = x^{-1}yx$, which we can rewrite as $yxy^{-n} = xy^{k-n}x$. Now substituting (ii) gives $y^{-(n-1)}xy = xy^{k-n}x$, which we may rewrite as $y^{n-k}x^{-1}y^{-(n-1)} = xy^{-1}x^{-1}$. Finally, substituting (iii) gives $y^{-(k+1)}x^{-1}y = xy^{-1}x^{-1}$, which is equivalent to $x = y^{-(k+1)}x^{-1}yxy = w_{k+1}$. $\quad\square$

**Theorem A.7.** *For each $n > 0$ and $k \in \mathbb{Z}$, $MS(n, w_k)$ is AC-equivalent to the presentation*

$$P(n, k) = \langle x, y \mid y^{n-k-1}x^{-1}yx = xyx^{-1}y^{n-k} \,,\; x = y^{-k}x^{-1}yxy\rangle,$$

*of total length $|k| + |n - k| + |n - k - 1| + 11$. For each fixed $n$, this length is minimum when $k = n - 1$, which gives the presentation*

$$P(n, n-1) = \langle x, y \mid x^{-1}yx = xyx^{-1}y \,,\; x = y^{-(n-1)}x^{-1}yxy\rangle,$$

*of length $n + 11$.*

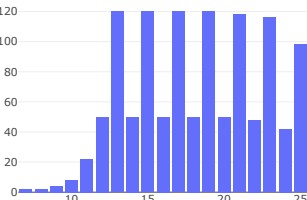

Figure 7: Number of presentations (Y-axis) versus $\ell$ (X-axis) for presentations in $\mathcal{D}$.

*Proof.* Starting with the presentation,

$$\text{MS}(n, w_k) = \langle x, y \mid x^{-1}y^n x = y^{n+1}, x = y^{-k}x^{-1}yxy \rangle,$$

the second relation may be equivalently expressed as

1. $y^k xy^{-1} = x^{-1}yx$

2. $y^{-1}xy^k = xyx^{-1}$

Multiplying the first relation by $y^{-1}x$ from the left and by $y^{-1}$ from the right, we get $y^{n-1}xy^{-1} = y^{-1}xy^n$, which we may rewrite as $y^{n-k-1}\left(y^k xy^{-1}\right) = \left(y^{-1}xy^k\right) y^{n-k}$. Substituting from (i) on the LHS and from (ii) on the RHS gives the required result. $\square$

Combining the previous three results shows that, for each fixed $n$, $\text{AK}(n)$ is AC-equivalent to all members of the infinite families $\text{MS}(n, w_k)$ and $P(n, k)$. From this, Theorem A.4 follows immediately from setting $k = n - 1$.

*Remark* A.8. While $P(n, n - 1)$ is always the shortest presentation from the $P$ family, other $P(n, k)$ may also give reductions in length of $\text{AK}(n)$. To find which presentations of $P$-family are shorter than $\text{AK}(n)$, we compare the length of $\text{AK}(n)$, i.e. $2n + 7$, with the length of $P(n, k)$ given by:

$$\begin{cases} 2n + 10 - k, & \text{if } k \leq n - 1, \\ 3k + 12 - 2n, & \text{if } k \geq n. \end{cases}$$

This is less than $2n + 7$ when $4 \leq k \leq n - 1$ or $n \leq k < \frac{1}{3}(4n - 5)$.

We also note that the AC-triviality of $\text{AK}(2)$, when combined with Theorem A.6, gives the following result.

**Corollary A.9.** *Each member of the infinite families $MS(2, w_k)$ and $P(2, k)$ is AC-trivial.*

## B    Benchmark Dataset

In this appendix, we describe the methodology used to obtain the benchmark dataset $\mathcal{D}$ in Section 4.1.

We start with a larger dataset of all presentations from the Miller–Schupp series $\text{MS}(n, w)$ with $n \leq 7$, length$(w) \leq 7$. This dataset contains AC-equivalent presentations which are related to each other through simple sequences of conjugation (AC3) moves. Consider, for example, the two presentations $\text{MS}(n, x^{-1}yx^{-1}y^2xy)$ and $\text{MS}(n, x^{-1}y^2xyx^{-1}y)$ for a fixed $n$. Conjugating the second relator in the first presentation by $x$ and $y^{-1}$ gives the second presentation. When evaluating the performance of an algorithm on the task of solving potential counterexamples, it is easy to obtain inflated numbers (say $80 - 90\%$) by solving presentations that are related by small sequences of AC-moves.

We make, therefore, the following reduction to our dataset: if $x^{-1}w$ and $x^{-1}w'$ are related through a sequence of conjugation (AC3) moves, keep only one of the two presentations $\text{MS}(n, x^{-1}w)$ and $\text{MS}(n, x^{-1}w')$. After this simplification, there are 170 choices of $x^{-1}w$ with $\ell(w) \leq 7$ as listed in the table below. With $n \leq 7$, our benchmark dataset $\mathcal{D}$ consists of $7 \times 170 = 1190$ presentations.

As discussed in Section 6.2, an intrinsic measure of hardness for a presentation is its length $\ell$. We plot the distribution of lengths in our dataset $\mathcal{D}$ in Figure 7.

In the table below, we specify the status of each presentation as "solved" or "unsolved" by our algorithms (cf. Section 5.2). 533 presentations solved by the greedy search are all mentioned in black; five presentations solved by a PPO agent with $T_{\max} = 200$ are mentioned in blue; and the two presentations solved by a PPO agent with piecewise-constant horizon length are mentioned in red. Lastly, presentations solved by a PPO agent trained with dynamic action spaces and supermoves (cf. Section 7) is shown in orange.

It may be easy to solve many more presentations in the dataset by simply scaling the techniques presented in this paper—for example, by using larger neural networks for PPO or by increasing the number of nodes visited during greedy search. However, what should be considered a true success for an algorithm is its ability to solve *hard* presentations, for which the shortest AC trivialization path may have thousands, if not hundreds of thousands, of moves. We expect that $\mathrm{MS}(n, y^{-1}x^{-1}yxy)$—a presentation known to be AC-equivalent to $\mathrm{AK}(n)$ (Myasnikov et al., 2002)—may be one such example.

| $w$ | $n=1$ | $n=2$ | $n=3$ | $n=4$ | $n=5$ | $n=6$ | $n=7$ |
|---|---|---|---|---|---|---|---|
| $y$ | solved | solved | solved | solved | solved | solved | solved |
| $y^{-1}$ | solved | solved | solved | solved | solved | solved | solved |
| $y^2$ | solved | solved | solved | solved | solved | solved | solved |
| $y^{-2}$ | solved | solved | solved | solved | solved | solved | solved |
| $y^3$ | solved | solved | solved | solved | solved | solved | solved |
| $y^{-3}$ | solved | solved | solved | solved | solved | solved | solved |
| $yxyx^{-1}$ | solved | solved | unsolved | unsolved | unsolved | unsolved | unsolved |
| $yxy^{-1}x^{-1}$ | solved | solved | solved | solved | solved | unsolved | unsolved |
| $y^4$ | solved | solved | solved | solved | solved | solved | solved |
| $x^{-1}y^{-1}xy$ | solved | solved | solved | solved | solved | unsolved | unsolved |
| $x^{-1}y^{-1}xy^{-1}$ | solved | solved | unsolved | unsolved | unsolved | unsolved | unsolved |
| $y^{-4}$ | solved | solved | solved | solved | solved | solved | solved |
| $yxy^2x^{-1}$ | solved | solved | solved | unsolved | unsolved | unsolved | unsolved |
| $yxyx^{-1}y$ | solved | solved | unsolved | unsolved | unsolved | unsolved | unsolved |
| $yxyx^{-1}y^{-1}$ | solved | solved | unsolved | unsolved | unsolved | unsolved | unsolved |
| $yxy^{-1}x^{-1}y$ | solved | solved | solved | solved | solved | unsolved | unsolved |
| $yxy^{-1}x^{-1}y^{-1}$ | solved | solved | solved | solved | solved | unsolved | unsolved |
| $yxy^{-2}x^{-1}$ | solved | solved | solved | unsolved | unsolved | unsolved | unsolved |
| $y^2xyx^{-1}$ | solved | solved | solved | unsolved | unsolved | unsolved | unsolved |
| $y^2xy^{-1}x^{-1}$ | solved | solved | unsolved | solved | unsolved | unsolved | unsolved |
| $y^5$ | solved | solved | solved | solved | solved | solved | solved |
| $yx^{-1}y^{-1}xy$ | solved | solved | solved | solved | solved | unsolved | unsolved |
| $yx^{-1}y^{-1}xy^{-1}$ | solved | solved | unsolved | unsolved | unsolved | unsolved | unsolved |
| $x^{-1}y^{-1}xy^2$ | solved | solved | solved | unsolved | unsolved | unsolved | unsolved |
| $x^{-1}y^{-1}xy^{-2}$ | solved | solved | solved | unsolved | unsolved | unsolved | unsolved |
| $x^{-1}y^{-2}xy$ | solved | solved | unsolved | solved | unsolved | unsolved | unsolved |
| $x^{-1}y^{-2}xy^{-1}$ | solved | solved | solved | solved | unsolved | unsolved | unsolved |
| $y^{-1}xyx^{-1}y^{-1}$ | solved | solved | solved | solved | solved | unsolved | unsolved |
| $y^{-1}xy^{-1}x^{-1}y^{-1}$ | solved | solved | unsolved | unsolved | unsolved | unsolved | unsolved |
| $y^{-5}$ | solved | solved | solved | solved | solved | solved | solved |
| $yx^2yx^{-2}$ | solved | unsolved | unsolved | unsolved | unsolved | unsolved | unsolved |
| $yx^2y^{-1}x^{-2}$ | solved | unsolved | unsolved | unsolved | unsolved | unsolved | unsolved |
| $yxy^3x^{-1}$ | solved | solved | solved | solved | solved | unsolved | unsolved |
| $yxy^2x^{-1}y$ | solved | solved | solved | unsolved | unsolved | unsolved | unsolved |
| $yxy^2x^{-1}y^{-1}$ | solved | solved | solved | unsolved | unsolved | unsolved | unsolved |
| $yxyx^{-1}y^2$ | solved | solved | unsolved | unsolved | unsolved | unsolved | unsolved |
| $yxyx^{-1}y^{-2}$ | solved | solved | unsolved | unsolved | unsolved | unsolved | unsolved |
| $yxy^{-1}x^{-1}y^2$ | solved | solved | solved | solved | unsolved | unsolved | unsolved |
| $yxy^{-1}x^{-1}y^{-2}$ | solved | solved | solved | solved | unsolved | unsolved | unsolved |
| $yxy^{-2}x^{-1}y$ | solved | solved | solved | unsolved | unsolved | unsolved | unsolved |

| $w$ | $n=1$ | $n=2$ | $n=3$ | $n=4$ | $n=5$ | $n=6$ | $n=7$ |
|---|---|---|---|---|---|---|---|
| $yxy^{-2}x^{-1}y^{-1}$ | solved | solved | solved | unsolved | unsolved | unsolved | unsolved |
| $yxy^{-3}x^{-1}$ | solved | solved | unsolved | unsolved | solved | unsolved | unsolved |
| $y^2xy^2x^{-1}$ | solved | solved | solved | solved | unsolved | unsolved | unsolved |
| $y^2xyx^{-1}y$ | solved | solved | solved | unsolved | unsolved | unsolved | unsolved |
| $y^2xyx^{-1}y^{-1}$ | solved | solved | solved | solved | unsolved | unsolved | unsolved |
| $y^2xy^{-1}x^{-1}y$ | solved | solved | unsolved | solved | unsolved | unsolved | unsolved |
| $y^2xy^{-1}x^{-1}y^{-1}$ | solved | solved | unsolved | solved | unsolved | unsolved | unsolved |
| $y^2xy^{-2}x^{-1}$ | solved | solved | solved | solved | solved | solved | solved |
| $y^3xyx^{-1}$ | solved | solved | solved | unsolved | unsolved | unsolved | unsolved |
| $y^3xy^{-1}x^{-1}$ | solved | solved | solved | unsolved | unsolved | solved | unsolved |
| $y^6$ | solved | solved | solved | solved | solved | solved | solved |
| $y^2x^{-1}y^{-1}xy$ | solved | solved | solved | solved | unsolved | unsolved | unsolved |
| $y^2x^{-1}y^{-1}xy^{-1}$ | solved | solved | unsolved | unsolved | unsolved | unsolved | unsolved |
| $yx^{-1}y^{-1}xy^2$ | solved | solved | solved | unsolved | unsolved | unsolved | unsolved |
| $yx^{-1}y^{-1}xy^{-2}$ | solved | solved | solved | unsolved | unsolved | unsolved | unsolved |
| $yx^{-1}y^{-2}xy$ | solved | solved | unsolved | solved | unsolved | unsolved | unsolved |
| $yx^{-1}y^{-2}xy^{-1}$ | solved | solved | solved | solved | unsolved | unsolved | unsolved |
| $x^{-2}y^{-1}x^2y$ | solved | unsolved | unsolved | unsolved | unsolved | unsolved | unsolved |
| $x^{-2}y^{-1}x^2y^{-1}$ | solved | unsolved | unsolved | unsolved | unsolved | unsolved | unsolved |
| $x^{-1}y^{-1}xy^3$ | solved | solved | unsolved | unsolved | solved | unsolved | solved |
| $x^{-1}y^{-1}xy^{-3}$ | solved | solved | solved | unsolved | solved | unsolved | unsolved |
| $x^{-1}y^{-2}xy^2$ | solved | solved | solved | solved | solved | solved | solved |
| $x^{-1}y^{-2}xy^{-2}$ | solved | solved | solved | solved | unsolved | unsolved | unsolved |
| $x^{-1}y^{-3}xy$ | solved | solved | solved | unsolved | unsolved | solved | unsolved |
| $x^{-1}y^{-3}xy^{-1}$ | solved | solved | solved | unsolved | unsolved | unsolved | unsolved |
| $y^{-1}xy^2x^{-1}y^{-1}$ | solved | solved | solved | unsolved | unsolved | unsolved | unsolved |
| $y^{-1}xyx^{-1}y^{-2}$ | solved | solved | solved | solved | unsolved | unsolved | unsolved |
| $y^{-1}xy^{-1}x^{-1}y^{-2}$ | solved | solved | unsolved | unsolved | unsolved | unsolved | unsolved |
| $y^{-1}xy^{-2}x^{-1}y^{-1}$ | solved | solved | solved | unsolved | unsolved | unsolved | unsolved |
| $y^{-1}x^{-1}y^{-2}xy$ | solved | solved | unsolved | solved | unsolved | unsolved | unsolved |
| $y^{-1}x^{-1}y^{-2}xy^{-1}$ | solved | solved | solved | solved | unsolved | unsolved | unsolved |
| $y^{-6}$ | solved | solved | solved | solved | solved | solved | solved |
| $yx^2y^2x^{-2}$ | solved | unsolved | unsolved | unsolved | unsolved | unsolved | unsolved |
| $yx^2yx^{-1}yx^{-1}$ | solved | unsolved | unsolved | unsolved | unsolved | unsolved | unsolved |
| $yx^2yx^{-2}y$ | solved | unsolved | unsolved | unsolved | unsolved | unsolved | unsolved |
| $yx^2yx^{-2}y^{-1}$ | solved | unsolved | unsolved | unsolved | unsolved | unsolved | unsolved |
| $yx^2yx^{-1}y^{-1}x^{-1}$ | solved | unsolved | unsolved | unsolved | unsolved | unsolved | unsolved |
| $yx^2y^{-1}x^{-1}yx^{-1}$ | solved | unsolved | unsolved | unsolved | unsolved | unsolved | unsolved |
| $yx^2y^{-1}x^{-2}y$ | solved | unsolved | unsolved | unsolved | unsolved | unsolved | unsolved |
| $yx^2y^{-1}x^{-2}y^{-1}$ | solved | unsolved | unsolved | unsolved | unsolved | unsolved | unsolved |
| $yx^2y^{-1}x^{-1}y^{-1}x^{-1}$ | solved | unsolved | unsolved | unsolved | unsolved | unsolved | unsolved |
| $yx^2y^{-2}x^{-2}$ | solved | unsolved | unsolved | unsolved | unsolved | unsolved | unsolved |
| $yxyxyx^{-2}$ | solved | unsolved | unsolved | unsolved | unsolved | unsolved | unsolved |
| $yxyxy^{-1}x^{-2}$ | solved | unsolved | unsolved | unsolved | unsolved | unsolved | unsolved |
| $yxy^4x^{-1}$ | solved | solved | solved | unsolved | unsolved | unsolved | unsolved |
| $yxy^3x^{-1}y$ | solved | solved | solved | unsolved | unsolved | unsolved | unsolved |
| $yxy^3x^{-1}y^{-1}$ | solved | solved | solved | unsolved | solved | unsolved | unsolved |
| $yxy^2x^{-1}y^2$ | solved | solved | solved | unsolved | unsolved | unsolved | unsolved |
| $yxy^2x^{-1}y^{-2}$ | solved | solved | solved | unsolved | unsolved | unsolved | unsolved |
| $yxyx^{-1}y^3$ | solved | solved | unsolved | unsolved | unsolved | unsolved | unsolved |
| $yxyx^{-1}y^{-3}$ | solved | solved | unsolved | unsolved | unsolved | unsolved | unsolved |
| $yxy^{-1}xyx^{-2}$ | solved | unsolved | unsolved | unsolved | unsolved | unsolved | unsolved |
| $yxy^{-1}xy^{-1}x^{-2}$ | solved | unsolved | unsolved | unsolved | unsolved | unsolved | unsolved |

| $w$ | $n=1$ | $n=2$ | $n=3$ | $n=4$ | $n=5$ | $n=6$ | $n=7$ |
|---|---|---|---|---|---|---|---|
| $yxy^{-1}x^{-1}y^3$ | solved | solved | solved | unsolved | unsolved | unsolved | unsolved |
| $yxy^{-1}x^{-1}y^{-3}$ | solved | solved | solved | solved | unsolved | unsolved | unsolved |
| $yxy^{-2}x^{-1}y^2$ | solved | solved | solved | unsolved | unsolved | unsolved | unsolved |
| $yxy^{-2}x^{-1}y^{-2}$ | solved | solved | solved | unsolved | unsolved | unsolved | unsolved |
| $yxy^{-3}x^{-1}y$ | solved | solved | unsolved | unsolved | solved | unsolved | unsolved |
| $yxy^{-3}x^{-1}y^{-1}$ | solved | solved | unsolved | unsolved | solved | unsolved | unsolved |
| $yxy^{-4}x^{-1}$ | solved | solved | solved | unsolved | unsolved | unsolved | unsolved |
| $y^2x^2yx^{-2}$ | solved | solved | unsolved | unsolved | unsolved | unsolved | unsolved |
| $y^2x^2y^{-1}x^{-2}$ | solved | solved | unsolved | unsolved | unsolved | unsolved | unsolved |
| $y^2xy^3x^{-1}$ | solved | solved | unsolved | solved | unsolved | unsolved | unsolved |
| $y^2xy^2x^{-1}y$ | solved | solved | solved | solved | unsolved | unsolved | unsolved |
| $y^2xy^2x^{-1}y^{-1}$ | solved | solved | solved | solved | unsolved | unsolved | unsolved |
| $y^2xyx^{-1}y^2$ | solved | solved | solved | unsolved | unsolved | unsolved | unsolved |
| $y^2xyx^{-1}y^{-2}$ | solved | solved | solved | solved | unsolved | unsolved | unsolved |
| $y^2xy^{-1}x^{-1}y^2$ | solved | solved | unsolved | solved | unsolved | unsolved | unsolved |
| $y^2xy^{-1}x^{-1}y^{-2}$ | solved | solved | unsolved | solved | unsolved | unsolved | unsolved |
| $y^2xy^{-2}x^{-1}y$ | solved | solved | solved | solved | solved | solved | unsolved |
| $y^2xy^{-2}x^{-1}y^{-1}$ | solved | solved | solved | solved | solved | solved | unsolved |
| $y^2xy^{-3}x^{-1}$ | solved | solved | solved | solved | solved | unsolved | unsolved |
| $y^3xy^2x^{-1}$ | solved | solved | solved | unsolved | unsolved | unsolved | unsolved |
| $y^3xyx^{-1}y$ | solved | solved | solved | unsolved | unsolved | unsolved | unsolved |
| $y^3xyx^{-1}y^{-1}$ | solved | solved | solved | unsolved | unsolved | unsolved | unsolved |
| $y^3xy^{-1}x^{-1}y$ | solved | solved | solved | unsolved | unsolved | unsolved | unsolved |
| $y^3xy^{-1}x^{-1}y^{-1}$ | solved | solved | solved | unsolved | unsolved | solved | unsolved |
| $y^3xy^{-2}x^{-1}$ | solved | solved | solved | unsolved | unsolved | solved | unsolved |
| $y^4xyx^{-1}$ | solved | solved | unsolved | solved | unsolved | unsolved | unsolved |
| $y^4xy^{-1}x^{-1}$ | solved | solved | solved | solved | unsolved | unsolved | unsolved |
| $y^7$ | solved | solved | solved | solved | solved | solved | solved |
| $y^3x^{-1}y^{-1}xy$ | solved | solved | solved | unsolved | unsolved | unsolved | unsolved |
| $y^3x^{-1}y^{-1}xy^{-1}$ | solved | solved | unsolved | unsolved | unsolved | unsolved | unsolved |
| $y^2x^{-1}y^{-1}xy^2$ | solved | solved | solved | unsolved | unsolved | unsolved | unsolved |
| $y^2x^{-1}y^{-1}xy^{-2}$ | solved | solved | solved | unsolved | unsolved | unsolved | unsolved |
| $y^2x^{-1}y^{-2}xy$ | solved | solved | unsolved | solved | unsolved | unsolved | unsolved |
| $y^2x^{-1}y^{-2}xy^{-1}$ | solved | solved | solved | solved | unsolved | unsolved | unsolved |
| $yx^{-2}y^{-1}x^2y$ | solved | unsolved | unsolved | unsolved | unsolved | unsolved | unsolved |
| $yx^{-2}y^{-1}x^2y^{-1}$ | solved | unsolved | unsolved | unsolved | unsolved | unsolved | unsolved |
| $yx^{-1}y^{-1}x^2yx^{-1}$ | solved | unsolved | unsolved | unsolved | unsolved | unsolved | unsolved |
| $yx^{-1}y^{-1}x^2y^{-1}x^{-1}$ | solved | unsolved | unsolved | unsolved | unsolved | unsolved | unsolved |
| $yx^{-1}y^{-1}xy^3$ | solved | solved | unsolved | unsolved | solved | unsolved | unsolved |
| $yx^{-1}y^{-1}xy^{-3}$ | solved | solved | solved | unsolved | solved | unsolved | unsolved |
| $yx^{-1}y^{-2}xy^2$ | solved | solved | solved | solved | solved | solved | unsolved |
| $yx^{-1}y^{-2}xy^{-2}$ | solved | solved | solved | solved | unsolved | unsolved | unsolved |
| $yx^{-1}y^{-3}xy$ | solved | solved | solved | unsolved | unsolved | solved | unsolved |
| $yx^{-1}y^{-3}xy^{-1}$ | solved | solved | solved | unsolved | unsolved | unsolved | unsolved |
| $x^{-2}y^{-1}x^2y^2$ | solved | unsolved | unsolved | unsolved | unsolved | unsolved | unsolved |
| $x^{-2}y^{-1}x^2y^{-2}$ | solved | unsolved | unsolved | unsolved | unsolved | unsolved | unsolved |
| $x^{-2}y^{-1}xyxy$ | solved | unsolved | unsolved | unsolved | unsolved | unsolved | unsolved |
| $x^{-2}y^{-1}xyxy^{-1}$ | solved | unsolved | unsolved | unsolved | unsolved | unsolved | unsolved |
| $x^{-2}y^{-1}xy^{-1}xy$ | solved | unsolved | unsolved | unsolved | unsolved | unsolved | unsolved |
| $x^{-2}y^{-1}xy^{-1}xy^{-1}$ | solved | unsolved | unsolved | unsolved | unsolved | unsolved | unsolved |
| $x^{-2}y^{-2}x^2y$ | solved | solved | unsolved | unsolved | unsolved | unsolved | unsolved |
| $x^{-2}y^{-2}x^2y^{-1}$ | solved | solved | unsolved | unsolved | unsolved | unsolved | unsolved |
| $x^{-1}y^{-1}x^2yx^{-1}y^{-1}$ | solved | unsolved | unsolved | unsolved | unsolved | unsolved | unsolved |

| $w$ | $n=1$ | $n=2$ | $n=3$ | $n=4$ | $n=5$ | $n=6$ | $n=7$ |
|---|---|---|---|---|---|---|---|
| $x^{-1}y^{-1}x^2y^{-1}x^{-1}y^{-1}$ | solved | unsolved | unsolved | unsolved | unsolved | unsolved | unsolved |
| $x^{-1}y^{-1}xy^4$ | solved | solved | solved | unsolved | unsolved | unsolved | unsolved |
| $x^{-1}y^{-1}xy^{-4}$ | solved | solved | solved | unsolved | unsolved | unsolved | unsolved |
| $x^{-1}y^{-1}x^{-1}y^{-1}x^2y$ | solved | unsolved | unsolved | unsolved | unsolved | unsolved | unsolved |
| $x^{-1}y^{-1}x^{-1}y^{-1}x^2y^{-1}$ | solved | unsolved | unsolved | unsolved | unsolved | unsolved | unsolved |
| $x^{-1}y^{-2}xy^3$ | solved | solved | solved | solved | solved | unsolved | unsolved |
| $x^{-1}y^{-2}xy^{-3}$ | solved | solved | unsolved | solved | solved | unsolved | unsolved |
| $x^{-1}y^{-3}xy^2$ | solved | solved | solved | unsolved | unsolved | solved | unsolved |
| $x^{-1}y^{-3}xy^{-2}$ | solved | solved | solved | unsolved | unsolved | unsolved | unsolved |
| $x^{-1}y^{-4}xy$ | solved | solved | solved | solved | unsolved | unsolved | unsolved |
| $x^{-1}y^{-4}xy^{-1}$ | solved | solved | unsolved | solved | unsolved | unsolved | unsolved |
| $y^{-1}xy^3x^{-1}y^{-1}$ | solved | solved | unsolved | unsolved | solved | unsolved | unsolved |
| $y^{-1}xy^2x^{-1}y^{-2}$ | solved | solved | solved | unsolved | unsolved | unsolved | unsolved |
| $y^{-1}xyx^{-1}y^{-3}$ | solved | solved | solved | unsolved | unsolved | unsolved | unsolved |
| $y^{-1}xy^{-1}x^{-1}y^{-3}$ | solved | solved | unsolved | unsolved | unsolved | unsolved | unsolved |
| $y^{-1}xy^{-2}x^{-1}y^{-2}$ | solved | solved | solved | unsolved | unsolved | unsolved | unsolved |
| $y^{-1}xy^{-3}x^{-1}y^{-1}$ | solved | solved | solved | unsolved | solved | unsolved | unsolved |
| $y^{-1}x^{-1}y^{-2}xy^2$ | solved | solved | solved | solved | solved | solved | unsolved |
| $y^{-1}x^{-1}y^{-2}xy^{-2}$ | solved | solved | solved | solved | unsolved | unsolved | unsolved |
| $y^{-1}x^{-1}y^{-3}xy$ | solved | solved | solved | unsolved | unsolved | unsolved | unsolved |
| $y^{-1}x^{-1}y^{-3}xy^{-1}$ | solved | solved | solved | unsolved | unsolved | unsolved | unsolved |
| $y^{-2}xyx^{-1}y^{-2}$ | solved | solved | unsolved | unsolved | unsolved | unsolved | unsolved |
| $y^{-2}xy^{-1}x^{-1}y^{-2}$ | solved | solved | solved | solved | unsolved | unsolved | unsolved |
| $y^{-7}$ | solved | solved | solved | solved | solved | solved | solved |

# C   Experimental Setup

In this appendix, we provide details of the experimental setup for various reinforcement learning agents discussed in the main text.

**PPO and A2C**

We trained agents using Synchronous Actor-Critic (A2C) (Mnih, 2016; Schulman et al., 2017) and Proximal Policy Optimization (PPO) (Schulman et al., 2017) algorithms. Both of these algorithms use an actor network to learn the policy function and a critic network to learn the value function. We used unshared networks in both cases. For the experiments of Section 5.1, all neural networks are chosen to be feed-forward neural networks with 2 hidden layers of 512 neurons. For the experiments of Section 5.2, both actor and critic networks are residual neural networks with six residual layers, where each residual block consists of two feed-forward layer with 512 neurons. We used Adam optimizer for training in all cases.

The performance of PPO is known to be highly sensitive to various implementation details in addition to the choice of hyperparameters (Huang et al., 2022a; Engstrom et al., 2019). Following (Engstrom et al., 2019), we used advantage normalization and clipped value loss. We also used reward normalization in all experiments. If the KL divergence between the old and the updated policy exceeded the target KL divergence in a mini-batch, we skipped the remaining mini-batches in the optimization phase to avoid a large jump in policy.

The A2C and PPO agents discussed in Section 5.1 are all trained for 100M environment interactions. Both algorithms achieve the best performance when trained with sparse rewards and AC action space. With this setup, their performance is compared in Figure 8.

The agents discussed in Section 5.2 are all trained for 1B environment interactions. The full list of hyperparameters for agents of Section 5.1 and Section 5.2 are given in Table 2 and Table 3 respectively.

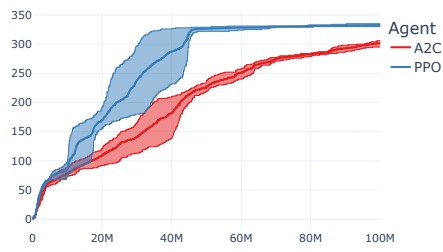

Figure 8: PPO vs. A2C

Table 2: Hyperparameters of PPO agents discussed in Section 5.1.

| Hyperparameter | Value |
| --- | --- |
| Rollout Length | 200 |
| Number of Parallel Actors | 28 |
| Total Number of Environment Interactions | $10^8$ |
| Learning rate | $1.0 \times 10^{-4}$ |
| Number of Epochs | 1 |
| Number of Mini-batches | 4 |
| Discount ($\gamma$) | 0.999 |
| GAE Parameter ($\lambda$) | 0.95 |
| Clipping Parameter ($\epsilon$) | 0.2 |
| Value Loss Coefficient ($c_1$) | 0.5 |
| Entropy Loss Coefficient ($c_2$) | 0.01 |
| Adam Epsilon Parameter | $10^{-5}$ |
| Target KL Divergence | 0.01 |

Our implementation of PPO is based on the CleanRL library (Huang et al., 2022b), which is distributed under the MIT License.

**DQN**

We trained our Deep Q-Network (DQN) agents using the Adam optimizer with a constant learning rate of 2.5e-4. We employed a replay buffer with a capacity of $500,000$ transitions, and training began

Table 3: Hyperparameters of PPO agents discussed in Section 5.2.

| Hyperparameter | Value |
| --- | --- |
| Rollout Length | 200 |
| Number of Parallel Actors | 1190 |
| Total Number of Environment Interactions | $10^9$ |
| Learning Rate | $10^{-4}$ |
| Number of Epochs | 3 |
| Number of Mini-batches | 4 |
| Discount ($\gamma$) | 0.999 |
| GAE Parameter ($\lambda$) | 0.95 |
| Clipping Parameter ($\epsilon$) | 0.2 |
| Value Loss Coefficient ($c_1$) | 0.5 |
| Entropy Loss Coefficient ($c_2$) | 0.01 |
| Adam Epsilon Parameter | $10^{-8}$ |
| Target KL Divergence | 0.01 |

after an initial $20,000$ timesteps of data collection. Updates were performed after every environment step by sampling 64 mini-batches; each of 256 transitions. For exploration, an epsilon-greedy strategy was used, with $\epsilon$ linearly annealed from 1.0 to 0.05 over the first 5 million timesteps. A target network was used for stability, with its weights being hard-copied from the online network every $1,000 updates$ ($\tau = 1.0$). The discount factor ($\gamma$) was set to 0.999. Each agent was trained for a total of 100 million timesteps across 1190 parallel environments.

### AlphaZero

We trained all AlphaZero agents for roughly 5.7M iterations using the AdamW optimizer with a learning rate of $1e - 4$. In each iteration, data was generated via self-play across 1190 parallel environments, with each episode lasting a maximum of 200 steps. At each step of self-play, a Monte Carlo Tree Search (MCTS) was conducted with 32 simulations to determine the next action. The search process was guided by the PUCT formula, with exploration parameters $c_1$ set to 1.25 and $c_2$ set to 19652. The training phase utilized mini-batches of 2048 samples, optimizing a combined loss function of softmax cross-entropy for the policy and mean squared error for the value. Value targets were computed as the undiscounted Monte Carlo returns from the self-play games.

### PPO with dynamic action spaces

The agent described in Section 7 is trained with the same hyperparameters as the ones described in Table 3 with a few small changes which we describe now. As in the table, the agent is trained for a total of 1 billion environment interactions. As we use 1190 parallel environments and each rollout consists of 200 timesteps, training consists of a total of $10^9/(1190 \times 200) \sim 4200$ iterations. We unmasked new actions every 100 iterations. Hence, the action space is modified 42 times. After each modification of the action space, we temporarily increased and then annealed the co-efficient of entropy loss from 0.01 to 0.05 over the next 20 epochs through a piecewise-linear schedule. This is to encourage the agent from exploring the state space with the help of new unmasked actions. By the end of training, a total of 405 actions were unmasked. Most of the unmasked actions corresponded to words of lengths 3 and 4.

## D   Path Analysis

In this appendix, we provide analysis of paths discovered by the agent of Section 7. As described there, the new agent, trained with a dynamic action space and supermoves, solved 495 presentations. 478 of these 495 presentations were also solved by greedy search, and 4 presentations were previously solved by the PPO agents described in Section 5.2. The 13 new presentations with their values of $n$, $w$ and path lengths (in terms of supermoves as well as elementary AC$'$ moves) are shown in Table 4.

| $n$ | $w$ | $\ell = 2n + 4 + \mathrm{Length}(w)$ | $N_{\text{supermoves}}$ | $N_{\text{AC' moves}}$ | $\ell$-**increase** |
|---|---|---|---|---|---|
| 5 | $y^{-1}xyx^{-1}y^{-1}$ | 19 | 196 | 284 | 12 |
| 5 | $yx^{-1}y^{-1}xy$ | 19 | 192 | 277 | 8 |
| 5 | $yxy^{-1}x^{-1}y$ | 19 | 182 | 248 | 11 |
| 5 | $yxy^{-1}x^{-1}y^{-1}$ | 19 | 172 | 231 | 14 |
| 4 | $y^{-1}xyx^{-1}y^{-2}$ | 18 | 156 | 210 | 21 |
| 4 | $yxy^{-1}x^{-1}y^2$ | 18 | 156 | 210 | 7 |
| 5 | $y^{-1}xy^3x^{-1}y^{-1}$ | 21 | 104 | 137 | 9 |
| 7 | $x^{-1}y^{-1}xy^4$ | 25 | 90 | 115 | 13 |
| 3 | $yxy^3x^{-1}y$ | 17 | 81 | 101 | 7 |
| 6 | $y^3xy^{-1}x^{-1}$ | 22 | 72 | 94 | 6 |
| 6 | $x^{-1}y^{-3}xy$ | 22 | 68 | 91 | 8 |
| 4 | $y^2xy^2x^{-1}y$ | 19 | 62 | 82 | 6 |
| 4 | $y^2xy^{-1}x^{-1}y^2$ | 19 | 53 | 74 | 7 |

Table 4: Path lengths and $l$-increase for various words $w$ and parameters $n$.

We observe that in 6 out of 13 cases, path lengths are less than our maximum horizon length of 200, but they exceed 200 when interpreted in terms of elementary AC' moves. In these cases, expanding

the action space has helped us discover AC-trivializations with paths longer than the maximum horizon length (200) of the original best agent from Section 5.2 as well.

# E Dependence of hardness measures on scale

The PPO agent analyzed in Section 6.2 was trained with $T_{\max} = 200$ for approximately 1 billion environment interactions. We observed that the lengths of trivialization paths discovered by the agent follow an almost continuous distribution, as shown in Figure 3b.

We hypothesize that at each scale of computation—measured by the total FLOPs used to train an agent—a similar distribution of path lengths emerges. A few presentations exhibit significantly long path lengths, making them the *hard* cases at that scale, while many others have short path lengths and can be considered *easy*. Here, we provide evidence supporting this hypothesis.

Keeping the actor and critic network sizes fixed at a three-layer feed-forward neural network with 512 neurons per hidden layer, we varied the total number of environment interactions used to train the PPO agent from 1 million to 80 million. For each fixed number of interactions, we trained agents with three different seeds and tuned the learning rate to maximize the number of solved presentations. The resulting scaling trend is shown on a log-log scale in Figure 9.

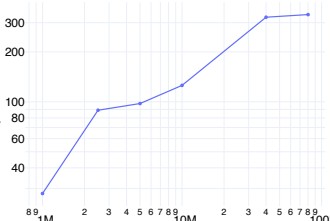

Figure 9: Number of presentations solved (Y-axis) against the total number of environment interactions used to train a PPO agent (X-axis).

In Figure 10, we plot the distributions of path lengths discovered by the most capable agent, which was trained for 80 million environment interactions. We color the bar of a presentation with respect to the first scale, in terms of total number of environment interactions, at which the presentation is solved. We make two observations about our results:

1. The overall distribution of path lengths closely resembles that of the agent trained with 1 billion environment interactions shown in Figure 3b.

2. Agents trained for longer amounts of time discover paths of longer lengths, indicating that agents trained for longer amounts of time are capable of solving *harder* presentations.

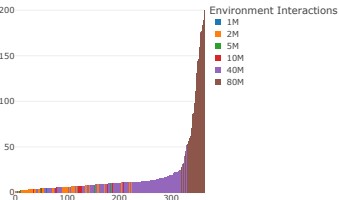

Figure 10: Bar plot of path lengths discovered by a PPO agent trained for 80 million environment interactions. Path length (Y-axis) against problem instances (X-axis). Bars are colored with respect to the minimum number of total environment interactions at which the presentation is first solved.

# F Equivalence of AC and AC′ Moves

In the main text of this paper, we defined AC-moves and AC′-moves. We will prove here that two presentations are AC-equivalent if and only if they are AC′-equivalent.

First, we recall the definitions of AC-moves:

1. Substitute some $r_i$ by $r_i r_j$ for $i \neq j$.
2. Replace some $r_i$ by $r_i^{-1}$.
3. Change some $r_i$ to $x_j^{\pm 1} r_i x_j^{\mp 1}$.

We also recall the definitions of AC′-moves:

1. Replace some $r_i$ by $r_i r_j^{\pm 1}$ for $i \neq j$.
2. Change some $r_i$ to $x_j^{\pm 1} r_i x_j^{\mp 1}$.

The difference between the two sets lies in how the inversion (AC2) of a relator is handled. We always follow an inversion (AC2) by a concatenation (AC1) in AC′1, while the original AC-moves allow for standalone inversion.

We can show that two presentations are AC-equivalent if and only if they are AC′-equivalent by proving that AC2 can be recovered from AC′-moves. To this end, consider a two-generator presentation $\langle x_1, x_2 \mid r_1, r_2 \rangle$. The sequence of AC′-moves,

$$r_2 \to r_2 r_1 \quad , \quad r_1 \to r_1 r_2^{-1} \quad , \quad r_2 \to r_2 r_1 \quad , \quad r_2 \to r_1 r_2 r_1^{-1},$$

results in the presentation $\langle x_1, x_2 \mid r_2^{-1}, r_1 \rangle$, which is the same as $r_2 \to r_2^{-1}$ up to swapping the two relators. A similar argument also holds for any pairs of relators in an $m$-relator presentation.

Strictly speaking, swapping two relators changes the presentation. However, we consider the two presentations $\langle x_1, x_2 \mid r_1, r_2 \rangle$ and $\langle x_1, x_2 \mid r_2, r_1 \rangle$ to be the same. When using AC′-moves to find AC-triviality of $m$-generator presentations, we enhance the notion of terminal states in reinforcement learning and goal states in classical search to include all presentations with $\ell = m$. When $m = 2$, there are eight such states:

$$\{ \langle x_1, x_2 \mid x_i^a, x_j^b \rangle \mid i, j = 1, 2; a, b = \pm 1; i \neq j \}$$

# G   A topological hardness measure for AC-trivialization problem

In Section 6, we studied hardness of presentations solved by our algorithms with respect to intrinsic as well as path-based measures. Here, we study a global hardness measure for the AC trivialization problem.

## G.1   Definition

A natural approach to define a global hardness measure for the AC graph is by aggregating the $\ell$-increase values across all presentations. However, this method suffers from overcounting: certain paths have an $\ell$-increase of 0, meaning that trivializing one endpoint automatically trivializes the other, incurring no additional cost.

We consider instead the following approach. Let $\Gamma_k$ be the induced subgraph containing all presentations with length at most $k$. Let us compare the components of $\Gamma_k$ and $\Gamma_{k+1}$.[5]

Consider a node $v$ of length $k+1$. If $v$ cannot be joined to a component of $\Gamma_k$ in $\Gamma_{k+1}$, then $v$ belongs to a new component that is said to be *born* at $k + 1$. If $v$ can be joined to a component of $\Gamma_k$, then its hardness needs not be considered since the path is of $\ell$-increase 0 and trivializing any presentation in the component with smaller length is more costly. If $v$ connects two distinct components, say born at $k_1 \leq k_2$, then the component born at $k_2$ is said to *die* at $k + 1$. By a similar reasoning, this component contributes $(k + 1) - k_2$ to the overall hardness of the problem, the difference between its death and birth values. If a component does not die, we set its death value to $\infty$.

We propose that the multiset consisting of all such pairs (birth, death) serves as a principled hardness measure for the AC trivialization problem and any other *path-to-the-base* search problem on a graph equipped with weights on its nodes. For any such problem, this multiset coincides with a key invariant

---

[5]Recall that the *components* of a graph refer to the equivalence classes of its nodes under the relation that identifies two if there is a path between them.

Table 5: Global hardness approximation for the AC- and AC′-trivialization problems. For each approximation depth $L$, $h_i$ represents the number of pairs $(b, d)$ in the hardness multiset satisfying $d - b = i$.

| | | **AC moves** | | | |
|---|---|---|---|---|---|
| $L$ | Nodes | Edges | $h_1$ | $h_2$ | $h_3$ |
| 11 | 350,356 | 1,002,439 | 4 | 0 | 0 |
| 12 | 791,140 | 2,251,375 | 16 | 0 | 0 |
| 13 | 3,238,052 | 9,321,629 | 72 | 4 | 0 |
| 14 | 7,199,908 | 20,573,343 | 144 | 4 | 0 |
| 15 | 29,243,812 | 84,391,763 | 508 | 52 | 8 |
| 16 | 64,623,652 | 185,162,236 | 1034 | 88 | 20 |

| | | **AC′ moves** | | | |
|---|---|---|---|---|---|
| $L$ | Nodes | Edges | $h_1$ | $h_2$ | $h_3$ |
| 11 | 350,356 | 655,928 | 19 | 0 | 0 |
| 12 | 791,140 | 1,467,080 | 67 | 0 | 0 |
| 13 | 3,238,052 | 6,107,112 | 243 | 16 | 0 |
| 14 | 7,199,908 | 13,414,744 | 483 | 16 | 0 |
| 15 | 29,243,812 | 55,306,744 | 1819 | 136 | 32 |
| 16 | 64,623,652 | 120,824,232 | 3923 | 208 | 80 |

from topological data analysis: the barcode of the persistent reduced homology in degree 0 of the associated filtered based graph.[6]

## G.2 Analysis

Let us now approximate the global $\ell$-increase hardness of the AC-trivialization problem. We will consider both sets of moves: AC and AC′.

We first need to specify a finite approximation to the problem. For $L \in \{11, \ldots, 16\}$, we take the induced subgraph containing all nodes admitting a trivialization of max length at most $L$. The upper bound was imposed by our computational resources, whereas the lower bound was chosen to highlight non-trivial behavior.

For each such approximation we compute its hardness multiset (as defined in Appendix G.1) using `giotto-TDA` (Tauzin et al., 2021)[7]. We then count the number $h_i$ of pairs $(b, d)$ with $d - b = i$. The results of our analysis for AC graph and AC′ graph are given in Table 5.

Taking the quotient of the values of both tables, we observe that, as expected, the number of nodes remains unchanged. Perhaps unsurprisingly, the number of edges scales consistently by approximately 1.5. More unexpectedly, we find that the scaling factor remains fairly constant across the other columns. This suggests that the overall complexity of the problem is preserved under the reformulation, albeit consistently scaled.

In the future, we aim to gather more data from other path-to-the-base search problems, such as those defined on the Reidemeister graph of knot presentations, and compare their hardness multisets systematically, using the bottleneck distance and other techniques from TDA.

In Appendix G.3, we investigate the extent to which similar topological invariants, computed on a filtered graph neighboring a presentation in $\mathcal{D}$, can predict its PPO-solved/unsolved label.

---

[6]We refer to any textbook on the subject for these concept, for example, Chapter 5 of Carlsson & Vejdemo-Johansson (2022).

[7]Specifically, its binding of the `SimplexTree` data structure introduced in Boissonnat & Maria (2014) and implemented in GUDHI (Maria et al., 2014).

Table 6: Quotient between corresponding values in the AC and AC′ tables in Table 5.

| $L$ | Nodes | Edges | $h_1$ | $h_2$ | $h_3$ |
|----|----|----|----|----|----|
| 11 | 1 | 1.5283 | 0.2105 | – | – |
| 12 | 1 | 1.5346 | 0.2388 | – | – |
| 13 | 1 | 1.5264 | 0.2963 | 0.2500 | – |
| 14 | 1 | 1.5336 | 0.2981 | 0.2500 | – |
| 15 | 1 | 1.5259 | 0.2793 | 0.3824 | 0.25 |
| 16 | 1 | 1.5325 | 0.2636 | 0.4231 | 0.25 |

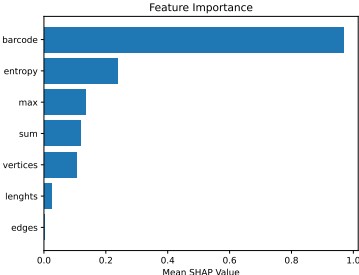

Figure 11: Feature importance measured by SHAP values for each feature, calculated in general setting, for the test set.

### G.3 Predicting hardness from local and topological features

We trained an XGBoost classifier on the data set $\mathcal{D}$ (Appendix B) to predict the label *PPO-(un)solved*, obtained by an RL agent. The data set consists of 417 PPO-solved and 773 PPO-unsolved presentations. The data set was randomly split into training and test subsets in a 4:1 ratio.

We use barcodes (defined at the end of Appendix G.2 as the numbers $h_i$ in the form of a vector) as the main source of local, topological features. The barcodes were calculated for the graph of a 5-step neighborhood, which is the set of presentations achievable using five or fewer AC-moves. The considered features include the number of vertices, the number of edges, the barcode vector, the maximal length of a bar, the sum of the barcode vector entries, persistence entropy, and basic features of the presentation, such as the lengths of each relator and the total length of the presentation.

We compared F1 scores on the test subset for classifiers trained with different feature subsets. Here is a brief summary. The length of the presentations, represented as three numbers (the lengths of each relator separately and their sum), achieved an F1 score of 0.885. The singular feature, the size of the neighborhood, achieved an F1 score of 0.930. Using the barcode vector as a single feature resulted in an F1 score of 0.943. An exhaustive search over all feature subsets enabled the selection of the optimal set of features based on the F1 score of the corresponding classifier. The selected subset includes the size of the neighborhood, its barcode vector, and the sum of the barcode entries. This classifier achieves an F1 score of 0.962.

We used the tree explainer technique and SHAP values to analyze the importance of each of considered features when building a modeled with all of them (number of vertices, number of edges, barcode vector, maximal length of a bar, sum of the barcode vector entries, persistence entropy, length of each relator, and length of the presentation). The lower entries of the barcode vector stood out as the most influential. Aggregated SHAP values are presented in the Figure 11. The feature 'barcode' inluces full barcode vector and the feature 'lengths' includes accumulated values of three features: length of each relator and length of the presentation.

