# OpenReview forum: "WHAT MAKES MATH PROBLEMS HARD FOR REINFORCEMENT LEARNING: A CASE STUDY"
_NeurIPS.cc/2025/Conference — NeurIPS 2025 poster_

### Official Review · Reviewer_zK73 · 2025-07-01

**Clarity:** 4
**Significance:** 3
**Originality:** 3
**Rating:** 5
**Confidence:** 3

**Summary:**

The paper presents a RL framework for finding trivialization paths of the Andrews–Curtis conjecture, a mathematical problem involving search in a large state space, but where solutions are easy to verify.
The authors define the problem as a RL environment with sparse rewards and train PPO and A2C agents to find solutions to some yet-unsolved instances. Comparing to two search benchmarks (breadth-first and greedy) the authors find the RL approach to be roughly equivalent, with RL solving a handful of problem instances that search methods failed to solve.
The authors analyze hardness measures for the problem and propose some ideas on how to tackle this problem for future study.

**Questions:**

- Can you present any results that indicate the new methods suggested in section 7 are useful? If you can, this would upgrade section 7 from a speculative discussion to a concrete contribution.

- I would suggest making a clearer separation between results related to deep learning and results that are not related to the field. In my opinion, the latter group of results should then be moved to the appendix, or published separately in a more relevant venue.

**Ethical Concerns:**

["NO or VERY MINOR ethics concerns only"]

**Final Justification:**

The authors provided new results using the techniques suggested in section 7, proving that these techniques are actually useful. Together with their promise of adding new RL benchmarks to the main results (DQN and AlphaZero), the paper is significantly improved in my opinion.

I do maintain some of my concern that the paper focuses too much on math novelties and not enough on RL related discoveries. If the paper is not accepted in its current form, I would suggest to the authors to revise it to better fit an ML/RL practitioners crowd, who care less about the problem solved and more about the method used to solve it.

**Limitations:**

yes

**Paper Formatting Concerns:**

No formatting concerns.

**Quality:**

3

**Strengths And Weaknesses:**

**Strengths**

1) Identifying the AC conjecture as a good test bed for RL, together with the dataset and clear problem definition, is a useful addition for the field.

2) The results show that the RL approach can solve a few instances where the benchmarks fail, even if only a few.

3) The paper is clearly written and easy to follow.

**Weaknesses**

1) The paper dedicates a significant portion to discussing the use case, but the scope of this paper is a bit small when looking only at the parts relevant to RL.
The novelties related to RL are: the cost function and environment definition, the performance analysis of PPO and A2C, hardness & path length distributions, and the dataset release.
On the other hand, things like the alternative AC' moves, theoretical results (5.3) and GS trivialization length analysis (6.2.3) might be novel, but are more of interest for mathematicians than for the RL/ML research community. Several parts of the main paper discuss use-case specific information or results obtained only using the search benchmarks. In a NeurIPS paper, I would expect the non-ML mathematical results to appear in the appendix rather than in the results section.

2) While the new algorithm suggestions in section 7 are indeed related to RL, they are not verified and are only presented as a promise of future study. These suggestions could very well be a crucial part of a future study, but in this paper they do not add much to the discussion without presenting an attempt to test them.
Section 7 is presented in the introduction as one of the paper's 3 main contributions, while in my opinion it is too speculative and would fit better as a note on future study in a 'Discussion'/'Conclusion' section, or in the appendix.


To conclude, my main concern about this paper is the relatively small amount of results that are relevant for RL practitioners.

---

> ### Author Rebuttal · Authors · 2025-07-31
>
> Dear reviewer,
>
> Thank you for your thoughtful and constructive review. We appreciate the time you took to engage with our work, in particular to note the analysis of the hardness measure, which is central to this work. We are also glad to hear you like the AC conjecture as a test bed for RL. We wish to return the favor and clarify your remaining concerns. (Please, do ask us for further clarifications if we fell short in this iteration.)
>
> While we agree that some parts of the paper are less focused on RL, this is primarily due to the fact that the paper is trying to accomplish a few different goals (which all revolve around RL, though).
>
> The first is to introduce the AC environment to the RL community, for its super-exponential hierarchy of hardness across the state space. As far as we know, such environments have *never* been discussed in the RL literature before. We are proud to be the first, and hope other RL researchers will find this type of new RL environments interesting.
>
> The second goal is our attempt to answer the question: *What would it take for an RL model to perform well in such types of environments with super-exponential hierarchy of sparse-reward long-horizon games?*
>
> Neither classical search algorithms nor any of the existing RL methods are suited for this challenging task. Therefore, one of our key messages in the paper was not so much the comparison of the performance between classical search algorithms and current RL tools, but rather the fact that neither performs well on these types of problems which are ubiquitous among long-standing research math problems.
>
> To underscore the novelty of the RL research required to address the above question, we should point out that at the present time we are not aware of any strong reason to believe that RL can solve such problems in the first place. In other words, the answer may simply be ``This is impossible for RL.''
>
> We hope this is not the case. And, the goal of the paper is to make first steps into RL research for such environments with super-exponential hierarchy of hardness. Specifically,
>
> - Multiple angles of attack --- classical search algorithms as well as existing RL algorithms --- for us are merely vehicles that led us to the notion of a new heuristic that we call ``hardness''.
> - We then study this notion of hardness from multiple vantage points and argue that an RL model must learn this hierarchy of scales of hardness in a dynamic way, through dynamic changes to the action space.
>
> In other words, all parts of our work are different ``supporting characters'' for the lead actor, the above mentioned question: What would it take for RL to learn how to find super-exponential sequences of actions required for a tiny fraction of the states?
>
> Based on your feedback, we realize the connection between different parts of the paper and the ``supporting'' roles of various parts could have been made stronger, something we will try and improve on in the camera-ready version of the work.
>
> Hopefully, this resolves one of your questions:
>
> *I would suggest making a clearer separation between results related to deep learning and results that are not related to the field. In my opinion, the latter group of results should then be moved to the appendix, or published separately in a more relevant venue.*
>
> To address the other question:
>
> *Can you present any results that indicate the new methods suggested in section 7 are useful?*
>
> First, we want to clarify that super-moves and dynamic action spaces are unavoidable, and the only question (which we do not attempt to address in this paper) is what novel RL algorithms implement these dynamic changes in the most optimal way. Secondly, below we present explicit examples of such implementations that show performance improvements in our AC environment; we will be happy to include these implementations into a camera-ready version of the paper or provide further details in this exchange.
>
> To start with a former, the key feature of the novel AC environment is that, unlike many well-studied RL environments, achieving the goal from some states requires a super-exponential number of steps compared to the polynomial number of steps for the majority of states. In fact, the number of such ``hard'' states is a tiny fraction of all states, which is why most of the standard RL models overfit on the generic states.
>
> The only way to convert the super-exponential length of the sequence of actions into a polynomial sequence is to allow the action space to admit new actions, that we call ``super-moves,'' which combine a number of actions from current iteration of the action space. For example, if one such super-move $S_1$ combines $N$ basic actions, then the next iteration may admit a new action $S_2$ that combines $O(N)$ such super-moves $S_1$, so that the total length of $S_2$ expressed in basic actions will be $\sim N^2$.  Repeating this process gives $|S_N| \sim N^N$ and allows to drastically reduce the super-exponential sequences of basic actions to only exponential or, with more efficient choices of super-moves, even further to just polynomial length sequences.
>
> Once we realize that super-moves are unavoidable in such exceptionally problems, the next question is how to best guide the RL model to make such dynamic changes. While we hope to open this question to a wider community of RL researchers, here we can offer concrete evidence based on a recent implementation. We have performed experiments with super-moves and dynamic action spaces, which show improvement over the baseline results reported in the manuscript.
>
> Our setup of dynamic action spaces looks as follows. We note that in many sequences of AC paths, a common strategy that the agent employs is to select several conjugation moves (AC$'$2) in a row followed by a single concatenation move (AC$'$1). We increased the size of the action space to include actions that perform conjugations by arbitrary words of generators and their inverses, followed by a single concatenation. This significantly enhances the size of the action space from the original $12$ to several thousands. In the beginning, however, we keep all of these new actions masked, letting the agent play the game with the original AC moves.
>
> As the agent solves any new presentation, we scan through the path it found, look for the sub-sequences of moves of the type described above and unmask only the actions corresponding to these sub-sequences. We repeat this process after every 100 rollout phases of the PPO training loop. With this approach, the agent solves $13$ new presentations from the Miller--Schupp dataset that were previously unsolved by greedy search and our reinforcement learning agents with static action spaces.
>
> These results shine a new light on the usefulness of dynamic action spaces and supermoves. We hope to add these results to Section 7 of the camera-ready version of the paper, along with providing paths for the $13$ newly solved presentations . We will also include the JAX code used for these experiments along with our submission.
>
> We also wish to remark that while techniques may seem extremely specific to the Andrews--Curtis environment at first, this is far from the truth. Many problems in mathematics come with examples that vary super-exponentially in hardness. Such problems arise in topology (e.g. in the problem of unknotting a knot), graph theory (e.g. in identifying the Ramsey numbers), and so on. If RL systems of the future are to help humanity solve these challenging problems, the introduction of supermoves and dynamic action spaces seems inevitable.

---

> > ### Comment · Reviewer_zK73 · 2025-08-01
> > **Rebuttal reply**
> >
> > Thank you for the detailed response.
> >
> > It is unfortunate that the new rebuttal rules don't allow you to post a draft of the new experimental results you claim you have. I am assuming you describe the new results in good faith, so I will base my final score on a future camera-ready version containing the results you promise.
> >
> > Even if your claim that (the use of) "super-moves and dynamic action spaces are unavoidable" is true, suggesting these techniques without validation is more fitting for a position paper than a conference paper. Proving these techniques improve performance is crucial, in my opinion.
> >
> > Could you elaborate more on the new results for section 7? You claim that the new method "solves 13 new presentations from the Miller--Schupp dataset" that plain PPO and search couldn't solve. What about the total number of presentations solved? How many presentations did plain PPO/search solve that the new method missed? Are these 13 newly solved presentations qualitatively different?
> >
> > I also think the results you promise in your response to other reviewers, adding analysis on DQN and AlphaZero, would boost the paper by increasing the amount of RL-related results.
> >
> > Another minor point regarding non-RL results: I still think things like section 5.3 don't belong in the main section of a NeurIPS paper, since they have no connection to RL/ML. You coming up with a mathematical theorem after being inspired by your work on ML is not what this conference is about.

---

> > > ### Author Response · Authors · 2025-08-05
> > > **Follow-up Response to Reviewer zk73**
> > >
> > > Dear reviewer,
> > >
> > > Thank you for your follow-up and further questions.
> > >
> > > We completely agree that including more RL-related results boosts the paper and makes it a better fit for the conference. We certainly commit to including the new results mentioned above as well as any code used to obtain the results in the camera ready copy. We also appreciate your feedback regarding section 5.3. In the camera-ready version, we are happy to replace this section by a one-line summary of the mathematical impact, especially as you suggest this will greatly improve the quality of the paper.
> > >
> > > ### New Experimental Results
> > >
> > > The new agent, trained with a dynamic action space and supermoves, solved 495 presentations (in contrast to 457 presentations solved by the best PPO agent described in Section 5.2). 478 of these 495 presentations were also solved by greedy search, and 4 presentations were previously solved by the PPO agents described in section 5.2. The 13 new presentations with their values of $n$, $w$ and path lengths (in terms of supermoves as well as elementary AC' moves) are given below.
> > >
> > > | $n$ | $w$      | $l = 2n + 4 + l(w)$ | Path length (in terms of supermoves) | Path length (in terms of AC' moves) | l-increase |
> > > |-----|----------|-------|------------|------------|---------|
> > > | 5   | YxyXY    | 19    | 196        | 284        |     12  |
> > > | 5   | yXYxy    |  19   | 192        | 277        |     8   |
> > > | 5   | yxYXy    | 19    | 182        | 248        |     11  |
> > > | 5   | yxYXY    | 19    | 172        | 231        |     14  |
> > > | 4   | YxyXYY   | 18    | 156        | 210        |     21  |
> > > | 4   | yxYXyy   | 18    | 156        | 210        |     7   |
> > > | 5   | YxYYYXY  | 21    | 104        | 137        |     9   |
> > > | 7   | XYxyyyy  | 25    | 90         | 115        |     13  |
> > > | 3   | yxyyyXy  | 17    | 81         | 101        |     7   |
> > > | 6   | yyyxYX   | 22    | 72         | 94         |     6   |
> > > | 6   | XYYYxy   | 22    | 68         | 91         |     8   |
> > > | 4   | yyxyyXy  | 19    | 62         | 82         |     6   |
> > > | 4   | yyxYXyy  | 19    | 53         | 74         |     7   |
> > >
> > >
> > > Here, $Y$ and $X$ denote $y^{-1}$ and $x^{-1}$ respectively.
> > >
> > > We trained this agent with roughly the same set of hyperparameters as the best PPO agent described in Section 5.2 (whose hyperparameters are given in Table 3 of Appendix C). More specifically, the maximum horizon length of the environment was kept at 200.
> > >
> > > The fourth column in the table above shows path lengths in terms of actions of the expanded action space. The fifth column gives the lengths of AC-trivialization paths after casting them in terms of elementary AC' moves.
> > >
> > > ### Analysis of Results
> > >
> > > We observe that in 6 out of 13 cases, path lengths are less than our maximum horizon length of 200, but they exceed 200 when interpreted in terms of elementary AC' moves. In these cases, expanding the action space has helped us discover AC-trivializations with paths longer than the maximum horizon length (200) of the original best agent from Section 5.2 as well. The other 7 cases do not exhibit the same trend. We do not yet understand what makes these presentations harder to solve for the original PPO agent as well as for greedy search and hence, leave this question as a future direction.
> > >
> > >
> > > ### Training Details for the New Agent
> > >
> > >
> > > We now provide some details of the experimental setup used to train the new agent. As remarked previously,  the entire action space consisted of 4 AC'1 moves along with conjugations of the form $r_i \to w r_i w^{-1}$ for $i=1, 2$ where $w$ is any word of length $1, \cdots, L$. $L$ is a hyperparameter, which we set to $6$ in our experiments. The number of choices of $w$ is $2 \times 3^L - 2$. Thus the action space has dimension $4 \times 3^L - 4 + 4 = 4 \times 3^L$. For $L=6$, this amounts to $2916$.
> > >
> > >
> > > The agent is trained for a total of 1 billion environment interactions. As we use 1190 parallel environments and each rollout consists of 200 timesteps, this consists of a total of $10^9 / (1190 \times 200) \sim 4200$ iterations. We unmasked new actions every 100 iterations. Hence, the action space was modified 42 times. After each modification of the action space, we temporarily increased and then annealed the co-efficient of entropy loss from 0.01 to 0.05 over the next 20 epochs through a piecewise-linear schedule. This is to encourage the agent from exploring the state space with the help of new unmasked actions.
> > >
> > > Initially, the action space had only 12 unmasked actions, corresponding to 12 elementary AC' moves. By the end of training, a total of 405 actions were unmasked. Most of the unmasked actions corresponded to words of lengths 3-4. We will add a histogram of distributions of lengths in the final manuscript.
> > >
> > >
> > > We reiterate our appreciation for your kind feedback, and we will be pleased to answer any more questions and include any more suggestions in our work.

---

> > > > ### Comment · Reviewer_zK73 · 2025-08-07
> > > >
> > > > Thank you for the detailed response, I think this makes section 7 a lot more valuable to the paper than the first draft. The fact some of the newly solved presentations have path lengths significantly longer than the horizon length shows your suggested  improvements have value. Regarding the new presentations that have a path length well below 200, maybe the original PPO couldn't find them simply due to how noisy exploration is? Maybe if you compare several PPO agent seeds you will also see that some of them fail to solve presentations other agents succeed in.
> > > >
> > > > Due to the new results in section 7 and the addition of new RL benchmarks, I will raise my score to a borderline accept (4). I agree with other reviewers that this paper is a bit too much on the math side, but the new additions provide enough RL-related novelty  for me to lean to accept.

---

### Official Review · Reviewer_Lzhs · 2025-07-02

**Clarity:** 3
**Significance:** 3
**Originality:** 4
**Rating:** 3
**Confidence:** 4

**Summary:**

"What makes math problems hard for reinforcement learning: a case study" is an interesting paper that provide a case study of using RL (mostly A2C and PPO agents) to address math problems. The exploration of the Andrews-Curtis conjecture as a reinforcement learning (RL) environment, combined with your mathematical contributions—such as resolving potential counterexamples in the Miller-Schupp series and demonstrating length reducibility in the Akhlutr-Kirby series—is intellectually compelling. The interdisciplinary approach, blending combinatorial group theory with RL, offers valuable insights into problem hardness and algorithmic challenges.

However, there are several reasons that I think this paper is not suitable for publication at NeurIPS 2025. The primary reason for this decision is that the paper's core contributions are more closely aligned with mathematical advancements in combinatorial group theory than with RL techniques, such as the use of A2C and PPO (Section 5). These are applied as tools to address these mathematical problems rather than introducing broadly applicable innovations in RL algorithms. While the authors argue that the introduction of "supermoves" (Section 6.23) might address the problem, it seems more like a conjecture at the moment that would require further study.

I believe this work would be exceptionally well-suited for a workshop or venue focused on the intersection of mathematics and machine learning, where the mathematical contributions and the application of RL to solve complex group-theoretic problems would be of primary interest.

**Questions:**

Can you elaborate more on how supermoves could be achieved?

**Ethical Concerns:**

["NO or VERY MINOR ethics concerns only"]

**Limitations:**

The paper's core contributions are more closely aligned with mathematical advancements in combinatorial group theory than with RL techniques, such as the use of A2C and PPO (Section 5). These are applied as tools to address these mathematical problems rather than introducing broadly applicable innovations in RL algorithms. While the authors argue that the introduction of "supermoves" (Section 6.23) might address the problem, it seems more like a conjecture at the moment that would require further study.

**Quality:**

3

**Strengths And Weaknesses:**

Stregnths -  Very interesting topics and the exploration of using RL to solve math problems is a promising direction
Weakness - More like a "case study" of a summary of a study which not yet at the point of being published as a conference paper.

---

> ### Author Rebuttal · Authors · 2025-07-31
>
> Dear reviewer,
>
> Thank you for taking the time to study our work. We are delighted to hear you found our RL exploration of the Andrews--Curtis conjecture combined with its mathematical contributions intellectually compelling. We are especially glad you appreciated the insights into hardness and algorithmic challenges. Hopefully we can alleviate some of the concerns you raised. From our perspective, your main concern seems to be:
>
>
> - *The paper's core contributions are more closely aligned with mathematical advancements in combinatorial group theory than with RL techniques, such as the use of A2C and PPO (Section 5). These are applied as tools to address these mathematical problems rather than introducing broadly applicable innovations in RL algorithms.*
>
>
> In particular, we see how one could reach the conclusion: *I believe this work would be exceptionally well-suited for a workshop or venue focused on the intersection of mathematics and machine learning, where the mathematical contributions and the application of RL to solve complex group-theoretic problems would be of primary interest.* The interdisciplinary math-ML workshops, though, focus on applications of ML to math problems, as you point out, or, in the opposite direction, on math applications to ML with a goal of either laying ML foundations or making ML more explainable. These workshops rarely tackle the problem of developing new RL techniques for which NeurIPS seems like a perfect venue.
>
> While we focused on the specific case of the Andrews-Curtis conjecture, many of the features which make RL challenging to apply to this problem are typical of most open mathematical research problems. It is these RL challenges that we hope to bring to the surface and discuss with experts like yourself at this NeurIPS meeting.
>
> The key feature of the AC-problem which (as far as we know) differentiates it from previously studied RL problems is that the gap between generic (average) case and the hardest case is not merely 10x or 100x, but rather many more orders of magnitude apart. This is why we believe the solution to this challenge can only come from RL community, and not from a typical math-ML workshop.
>
> Past RL success stories are, in a combinatorial sense, easier than the challenge described here: in board games like Go and Chess, even hand-made puzzles never require more than $100$ steps of foresight to solve. The 10-year history of achieving expert human performance on the infamous ``Montezuma's Revenge" also falls far short of the challenge above, and so are many RL environments that involve autonomous vehicles, etc.
>
> The Andrews-Curtis environment of this paper is a new kind of game, where an RL agent *must* make a super-exponential number of moves in a small number of important states. The question we tackle is: How can it do this?
>
> The only way to turn the super-exponential number of actions to a sequence of exponential or polynomial length is to allow the action space to be dynamic. Admitting to it new types of moves that progressively combine large chunks of actions from the previous iteration of the action space.
>
> This brings us to your companion remark:
>
> - *While the authors argue that the introduction of ``supermoves" (Section 6.23) might address the problem, it seems more like a conjecture at the moment that would require further study.*
>
> We want to clarify that the use of super-moves is unavoidable, it is a must in these kind of games where the distribution of hard and easy instances is extremely non-uniform and spans many orders of magnitude. The main challenge then lies in the algorithmic design that implements action changes in the most optimal case. We realize we should have stressed this point more in the paper, and will be happy to do so in the camera-ready version.
>
> In other words, one of the main points of our paper is that *none* of the traditional RL techniques can handle environments with super-exponential hierarchy of hardness of the state space. While we discussed the performance of only policy-gradient methods in the manuscript, we have also experimented with off-policy and search-based algorithms. In all cases, these algorithms struggle with the extremely hard set of presentations. (We will include the performance levels of these algorithms in Section 4 of the camera-ready version of the paper.) We hope this aspect alone merits sufficient novelty for the RL community.
>
> And, as for concrete experiments with dynamic action spaces, we also want to share some details to elevate this from the conceptual --- and, in our opinion, important --- discussion to practical study. We have performed some experiments with super-moves and dynamic action spaces, which have shown improvement over the baseline results reported in the manuscript.
>
> Specifically, our setup of dynamic action spaces looks as follows. We note that in many sequences of AC paths, a common strategy that the agent employs is to select several conjugation moves (AC$'$2) in a row followed by a single concatenation move (AC$'$1). We increased the size of the action space to include actions that perform conjugations by arbitrary words of generators and their inverses, followed by a single concatenation. This significantly enhances the size of the action space from the original $12$ to several thousands. In the beginning, however, we keep all of these new actions masked, letting the agent play the game with the original AC moves.
>
> As the agent solves any new presentation, we scan through the path it found, look for the sub-sequences of moves of the type described above and unmask only the actions corresponding to these sub-sequences. We repeat this process after every 100 rollout phases of the PPO training loop. With this approach, the agent solves $13$ new presentations from the Miller--Schupp dataset that were previously unsolved by greedy search and our reinforcement learning agents with static action spaces.
>
> These results provide some empirical evidence for the usefulness of dynamic action spaces and supermoves. We intend to add these results to Section 7 of the camera-ready version of the paper, along with providing paths for the $13$ newly solved presentations . We will also include the JAX code used for these experiments along with our submission.
>
> Hopefully we managed to address all primary concerns. If not, please ask us for further clarifications.

---

> > ### Comment · Reviewer_Lzhs · 2025-08-07
> >
> > Thanks the authors for the response. While the argument make sense, I still failed to see concrete evidence of the advancement of either tackling the AC conjecture or how the authors would come up with supermoves. To me this still looks like an interesting case study that are just not ready for formal publication yet.

---

### Official Review · Reviewer_eKhC · 2025-07-03

**Clarity:** 3
**Significance:** 3
**Originality:** 3
**Rating:** 5
**Confidence:** 3

**Summary:**

This paper studies why reinforcement learning (RL) struggles with solving certain mathematical problems by analyzing the Andrews–Curtis (AC) conjecture—a long-standing problem in group theory. The authors frame the AC trivialization task as a long-horizon, sparse-reward RL environment with an intrinsic distribution of problem hardness. They analyze how RL agents like PPO perform compared to classical algorithms, and identify limitations in greedy approaches. Notably, their methods solve previously open instances in the Miller–Schupp and Akbulut–Kirby series, proving new infinite families are AC-trivial. Based on these insights, they propose novel RL techniques using supermoves and adaptive action spaces to better tackle hard problem instances.

**Questions:**

1. In Theorem A.2, you proved that $\text{MS}(1, w)$ is AC-trivial for all $w$ by applying substitution and conjugation identities derived from the relators. Can you formalize the conditions under which these substitution steps are guaranteed to terminate for arbitrary $w$, especially given that $w$ only needs to have zero exponent sum in $x$? Is there a provable upper bound on the number of AC-moves required?

2. Theorem A.4 showed that for all $n \geq 2$, $\text{AK}(n)$ is AC-equivalent to a length-$n + 11$ presentation
  $\langle x, y \mid x^{-1}yx = xyx^{-1}y,\ x y^{n-1} x = y x y \rangle$. Can you clarify whether this equivalence is minimal in the number of AC-moves required, or just in presentation length? Additionally, is this reduction provably optimal in terms of minimal length among all AC-equivalent presentations?

**Ethical Concerns:**

["NO or VERY MINOR ethics concerns only"]

**Final Justification:**

The authors’ openness to refining their work in response to reviewer feedback, coupled with the new experiments they conducted and shared during the rebuttal, leads me to believe this paper merits an opportunity for acceptance, and I am inclined to place trust in their ability to deliver a strong final version.

**Limitations:**

yes

**Quality:**

3

**Strengths And Weaknesses:**

Strengths
- Frames the Andrews–Curtis trivialization problem as a long-horizon, sparse-reward RL task and evaluates PPO vs A2C with controlled variations in reward and horizon schedules.
- Classical baselines (BFS, greedy search) are implemented fairly, enabling a rigorous comparison showing RL solves harder, non-monotonic paths.
- RL-guided exploration leads to new mathematical results, including full AC-triviality proofs for $\text{MS}(1, w)$, $\text{MS}(n, y^{-1} x y x^{-1})$, and length reduction in $\text{AK}(n)$.

Weaknesses
- Only on-policy methods (PPO, A2C) are tested; no search-guided or off-policy RL approaches are explored.
- Supermoves and adaptive action spaces are proposed but not empirically validated.
- Problem scale is limited to short presentations ($n \leq 7$), leaving generalization to harder instances uncertain.

---

> ### Author Rebuttal · Authors · 2025-07-31
>
> Dear reviewer,
>
> Thank you for your thoughtful and constructive review. We appreciate the time you took to engage with our work and think you raised some interesting questions and concerns regarding potential areas of improvement of our paper.
>
> 1. We begin by clarifying a potential misunderstanding, and which we have now taken care of to prevent in the revised manuscript. In the weaknesses section, you write:
>
> *Problem scale is limited to short presentations ($n \leq 7$), leaving generalization to harder instances uncertain.*
>
> This likely arises from a miscommunication regarding the role of the parameter $n$ in Miller--Schupp presentations. Specifically, $n$ is not the length of a presentation but rather appears in the exponent of a relator. Our dataset---up to certain trivial identifications described in Appendix~D---contains all presentations of the form
>
> $$MS(n, w) = \langle x, y \mid x^{-1} y^n x = y^{n+1},\ x = w \rangle, $$
>
> with $n \leq 7$ and $\mathrm{length}(w) \leq 7$. The length of the presentation $MS(n, w)$ is $2n + 4 + \mathrm{length}(w)$, which is greater than $12$ for 92% of the presentations in the dataset. The distribution of lengths is shown in Figure 7 in the appendix.
>
> The importance of having a large number of presentations with lengths greater than $12$ is that the conjecture is known to hold true for any presentation with length less than or equal to $12$. In this paper, we wanted our dataset to be in the regime where we use reinforcement learning to find new results in mathematics. For most of these presentations, a trivialization found by purely mathematical techniques would be recognized as a valuable mathematical result. Here, we have managed to do so using classical search and reinforcement learning techniques.
>
> To preempt this potential misunderstanding, we have included the following clarifying sentences to Section 4.1:
>
> *This dataset contains Miller–Schupp presentations, $MS(n, w)$, with  $n \leq 7$ and $\text{length}(w) \leq 7$, and includes a total of 1190 presentations. The lengths of these presentations lie in the range $[7, 25]$ and, for the vast majority of them, their AC-triviality was previously unknown.*
>
>
> We observe in Figure 2(a) that as the length of presentation increases, the efficacy of classical search as well as reinforcement learning algorithms declines significantly. We experimented with presentations of even longer lengths and observed that this trend continues, which indicates the inherent difficulty of solving such presentations. If you suggest, we will be happy to include more presentations in the camera-ready version of the paper, significantly expanding the values of $n$ and $\mathrm{length}(w)$.
>
> 2. We now turn our attention to the following remark.
>
> *Only on-policy methods (PPO, A2C) are tested; no search-guided or off-policy RL approaches are explored.*
>
> We appreciate your feedback. Indeed, in the manuscript, we only discuss on-policy methods, which is an oversight on our part. We have experimented with an off-policy RL algorithm (DQN with soft target network updates) and a search-guided method (AlphaZero) during the rebuttal period.
>
> We observed that a well-tuned DQN under-performs compared to PPO, solving only 150 presentations from our dataset, when trained for 100 million environment interactions. In contrast, the best PPO agents solved 300+ presentations for the same amount of training period, as shown in Figure 1(a) of the paper.
>
> AlphaZero performs search-based reinforcement learning, and hence, training it for 100 million environment interactions does not provide a fair comparison to PPO. For the purpose of comparing it with PPO, we trained for equal amount of wall-clock time, i.e. 24 hours. We observed that the performance of a well-tuned AlphaZero agent is comparable to PPO, and hence, it can serve as a good alternative for further experimentation and scaling of the model.
>
> We have edited Sections 4 and 5 in the manuscript to include the comparison of these algorithms with policy-gradient algorithms. We will also include the code used to run these experiments along with the camera-ready version of the paper.
>
> 3. We will now address your feedback regarding the contents of Section 7.
>
> *Supermoves and adaptive action spaces are proposed but not empirically validated.*
>
> During the rebuttal period, we have performed new experiments with super-moves and dynamic action spaces, which show improvement over the baseline results reported in the manuscript.
>
> Our setup of dynamic action spaces looks as follows. We note that in many sequences of AC paths, a common strategy that the agent employs is to select several conjugation moves (AC'2) in a row followed by a single concatenation move (AC'1). We increased the size of the action space to include actions that perform conjugations by arbitrary words of generators and their inverses, followed by a single concatenation. This significantly enhances the size of the action space from the original $12$ to several thousands. In the beginning, however, we keep all of these new actions masked, letting the agent play the game with the original AC moves.
>
> As the agent solves any new presentation, we scan through the path it found, look for the sub-sequences of moves of the type described above and unmask only the actions corresponding to these sub-sequences. We repeat this process after every 100 rollout phases of the PPO training loop. With this approach, the agent solves $13$ new presentations from the Miller--Schupp dataset that were previously unsolved by greedy search and our reinforcement learning agents with static action spaces.
>
> These results shine a new light on the usefulness of dynamic action spaces and supermoves. We commit to adding these results to Section 7 of the camera-ready version of the paper, along with providing paths for the $13$ newly solved presentations. We will also include the JAX code used for these experiments along with our submission.
>
> 4. We now turn our attention to your questions.
>
> - *In Theorem A.2, you proved that ${MS}(1, w)$ is AC-trivial for all $w$ by applying substitution and conjugation identities derived from the relators. Can you formalize the conditions under which these substitution steps are guaranteed to terminate for arbitrary $w$, especially given that $w$ only needs to have zero exponent sum in $x$? Is there a provable upper bound on the number of AC-moves required?*
>
> These substitution steps are guaranteed to terminate for arbitrary $w$. Indeed, for each $x$, the proof moves it past each $y$ and $y^{-1}$ that appear to the left of it, and for each $x^{-1}$, the proof moves it past each $y$ and $y^{-1}$ that appear to the right of it. Then we conjugate by $x^{-1}$, each time canceling an $x$ and $x^{-1}$ pair. As there are finitely many $x^{\pm 1}$ and $y^{\pm 1}$ appearances, this will always terminate.
>
> A rough upper bound on the number of AC-moves required in the substitution algorithm described is $|w|2^{|w|}$ where $|w|$ is the length of $w$. To see this, we note that the first $x^{\pm 1}$ requires at most $|w|$ moves to get it to the correct position. Whenever it passes a $y^{\pm 1}$, we replace that by $y^{\pm 2}$ and so the second $x^{\pm 1}$ requires at most $2|w|$ moves to get it to the right position. Continuing on, the $i^\text{th}$ $x^{\pm 1}$ moved requires at most $2^{i - 1}|w|$ moves and so, as the number of $x^{\pm 1}$s is at most $|w|$, the number of moves required is less than $(1 + 2 + \cdots + 2^{|w| - 1})|w| < |w|2^{|w|}$. In practice the true upper bound is much smaller than this but this suffices for showing that an upper bound exists.
>
> - *Theorem A.4 showed that for all $n \geq 2$, ${AK}(n)$ is AC-equivalent to a length $n + 11$ presentation $\langle x, y \mid x^{-1}yx = xyx^{-1}y,\ x y^{n-1} x = y x y \rangle$. Can you clarify whether this equivalence is minimal in the number of AC-moves required, or just in presentation length? Additionally, is this reduction provably optimal in terms of minimal length among all AC-equivalent presentations?*
>
> The equivalence of $\text{AK}(n)$, which is of length $2n+7$, with the length $n+11$ presentation $\langle x, y \mid x^{-1}yx = xyx^{-1}y,\ x y^{n-1} x = y x y \rangle$, is the shortest presentation currently known to be AC-equivalent to $\text{AK}(n)$. In other words, this is the ``current best," and moreover is the first example of any length reduction for $n \geq 3$.
>
> It is possible that there exist even shorter presentations that are AC-equivalent to $\text{AK}(n)$ (if the AC conjecture is true, they are AC-equivalent to presentations of length 2). However, we would like to take this opportunity to emphasize the importance of our result: $\text{AK}(n)$ were introduced as potential counterexamples of AC conjecture more than 4 decades ago. A lot of research had been done on these presentations, and it was unknown before our paper whether they admit length reduction at all. Our results create a new opening towards finding solutions for these presentations by showing that a significant reduction in length is possible.
>
> We did not investigate whether the AC-equivalence we find is minimal in terms of the number of AC-moves required, and we do not expect them to be. We are primarily interested in the existence of any AC-triviality paths as opposed to finding the most optimal ones, especially in the cases of long-standing potential counterexamples of the Akbulut--Kirby series.

---

### Official Review · Reviewer_nQDy · 2025-07-03

**Clarity:** 2
**Significance:** 1
**Originality:** 2
**Rating:** 3
**Confidence:** 1

**Summary:**

This paper studies using RL for resolving or disproving math conjectures (e.g., by finding counterexamples). In particular they study the Andrews-Curtis conjecture from group theory. They focus on the case with 2 variables and length 1,..,7. They show that specific RL approaches can confirm some instances of the conjecture, which were known to be true, in roughly 200 out of 1000 cases.

Based on their first empirical results they propose some ideas/directions for future work.

**Questions:**

.

**Ethical Concerns:**

["NO or VERY MINOR ethics concerns only"]

**Final Justification:**

The authors clarified the setup and contributions. This makes indeed the paper more interesting and stronger, and I slightly raised my score. However the contributions were not communicated well in the original submission. Also I cannot judge added experiments concerning additional techniques, which were only discussed during the rebuttal. I recommend to fully rewrite the paper and then submit it to the next conference.

**Limitations:**

Limitations are addressed but see points above.

**Paper Formatting Concerns:**

.

**Quality:**

1

**Strengths And Weaknesses:**

The studied problem is important and in the long run RL methods could become an important tool for resolving math conjectures. The tackled conjecture seems to yield an interesting benchmark to track progress on the usefulness of RL for proving/disproving conjcetures. However this study has various limitations

Weaknesses:
* Limited scope. The conjecture is tested only for two variables and length up to 7. Previous work already confirmed the conjecture for length up to 13. Even in this limited scenario the proposed approaches can only confirm roughly 20% of the instances.
* Besides the experimental results the paper is rather hypothetical and conceptual (as the authors themselves admit in the limitations). It is unclear if these proposals will actually lead to further improvements.

---

> ### Author Rebuttal · Authors · 2025-07-31
>
> Dear reviewer,
>
> We appreciate the time you took to engage with our work and for highlighting both its strengths and areas for improvement. Below, we respond to your comments and clarify the points you raised.
>
> 1. We note in your summary the following statement:
>
> *They focus on the case with 2 variables and length $1,\dots,7$. Previous work already confirmed the conjecture for length up to 13.*
>
> This likely arises from a miscommunication regarding the role of the parameter $n$ in Miller--Schupp presentations. Specifically, $n$ is not the length of a presentation but rather appears in the exponent of a relator. Our dataset---up to certain trivial identifications described in Appendix~D---contains all presentations of the form
>
> $$MS(n, w) = \langle x, y \mid x^{-1} y^n x = y^{n+1},\ x = w \rangle,$$
>
> with $n \leq 7$ and $\mathrm{length}(w) \leq 7$. The length of the presentation $MS(n, w)$ is $2n + 4 + \mathrm{length}(w)$, which is greater than $12$ for 92% of the presentations in the dataset. The distribution of lengths is shown in Figure 7 in the appendix.
>
> That a large number of presentations have lengths greater than $12$ is quite meaningful to us. As you commented, the conjecture is known to hold true for all presentations with length less than or equal to $12$. We wanted our dataset to be in the regime where it largely consists of previously-open problems in the field, and finding a trivialization for any of these presentations would be recognized as a valuable mathematical result. Here, we have managed to do so using classical search and reinforcement learning techniques.
>
> To preempt this potential misunderstanding, we have included the following clarifying sentences to Section 4.1:
>
> *This dataset contains Miller–Schupp presentations, $MS(n, w)$, with  $n \leq 7$ and $\text{length}(w) \leq 7$, and includes a total of 1190 presentations. The lengths of these presentations lie in the range $[7, 25]$ and, for the vast majority of them, their AC-triviality was previously unknown.*
>
> We also note that while we only discuss the case of $m=2$ generators in the paper, this was mainly to keep the paper as focused as possible. In our experiments, we also studied a family of presentations with $m=3$, which is of interest in mathematics. Here, we also solved presentations that are previously open. If you suggest, we will be able to include these results in an appendix in the camera-ready version of the paper. Below, we have written a small rough draft of this new addition.
>
>
> **Balanced presentations with 3 generators**
>
> In addition to the well-known Akbulut–Kirby and Miller–Schupp families in the case of two-generators, we also explored the question of AC-triviality for a 3-parameter family of presentations.
>
> $$P \left( 2k+1, 2; \frac{2n}{2nm+1} \right) = \left\langle x, y \mid x^{2k+1} = y^2, y x^{-k} \left( (y x^{-k})^n (y^{-1} x^k)^n \right)^m \right\rangle$$
>
> This family has a geometric origin, with the presentations being trivial implied by the corresponding R-links having Generalized Property R \cite{Gompf, Robert E. and Scharlemann, Martin and Thompson, Abigail}.
> Our reinforcement learning agents trivialized the following presentation, which was previously a potential counterexample to Andrews--Curtis conjecture.
>
> $$P (5,2; \tfrac{4}{5} ) = \langle x,y \mid x^5 = y^2, y x^{-2} (y x^{-2})^2 (y^{-1} x^2)^2 \rangle$$
>
> The discovered AC path is of length $49$:
>
> $$ h_{11} \cdot h_{11} \cdot h_{1} \cdot h_{11} \cdot h_{11} \cdot h_{11} \cdot h_{6} \cdot h_{8} \cdot h_{8} \cdot h_{6} \cdot $$
> $$ h_{0} \cdot h_{8} \cdot h_{8} \cdot h_{9} \cdot h_{11} \cdot h_{1} \cdot h_{9} \cdot h_{11} $$
> $$ h_{5} \cdot h_{7} \cdot h_{1} \cdot h_{9} \cdot h_{11} \cdot h_{8} \cdot h_{3} \cdot h_{5} \cdot h_{4} \cdot h_{4} \cdot h_{2} $$
> $$\cdot h_{8} \cdot h_{10} \cdot h_{4} \cdot h_{9} \cdot h_{0} \cdot h_{5} \cdot h_{4} $$
> $$ h_{4} \cdot h_{1} \cdot h_{9} \cdot h_{10} \cdot h_{4} \cdot h_{3} \cdot h_{7} \cdot h_{5} \cdot h_{2} \cdot h_{2} \cdot h_{1} \cdot h_{5} \cdot h_{1} $$
>
> The sequence should be read from left to right, e.g. we first apply $h_{11}$ twice, then $h_{1}$, and so on. $h_i$ are AC$'$-moves defined as follows.
>
> $$ h_1 = \ r_2 \rightarrow r_2 r_1,  \quad h_5 = \ r_2 \rightarrow x^{-1} r_2 x, \quad h_9 = \ r_2 \rightarrow x r_2 x^{-1}, $$
> $$h_2 = \ r_1 \rightarrow r_1 r_2^{-1},  \quad h_6 = \ r_1 \rightarrow y^{-1} r_1 y,  \quad h_{10} = \ r_1 \rightarrow y r_1 y^{-1}, $$
> $$ h_3 = \ r_2 \rightarrow r_2 r_1^{-1},  \quad h_7 = \ r_2 \rightarrow y^{-1} r_2 y,  \quad h_{11} = \ r_2 \rightarrow y r_2 y^{-1}, $$
> $$h_4 = \ r_1 \rightarrow r_1 r_2,  \quad h_8 = \ r_1 \rightarrow x r_1 x^{-1},  \quad h_{12} = \ r_1 \rightarrow x^{-1} r_1 x, $$
>
> 2. We now turn our attention to the following statement.
>
> *Even in this limited scenario the proposed approaches can only confirm roughly 20\% of the instances.*
>
> As we described above, most of the presentations in our benchmark dataset are of length greater than 12, and were previously unknown to be solvable. Thus, we are already in the regime of solving *long-standing open research-level* problems in mathematics. From our dataset of 1190 presentations, we solved approximately 600, which is close to 50%. In addition, we used our empirical results to solved certain *infinite* subfamilies as described in Section 5.3 and Appendix A.
>
> 3. The second weakness listed is the following:
>
> *Besides the experimental results the paper is rather hypothetical and conceptual (as the authors themselves admit in the limitations). It is unclear if these proposals will actually lead to further improvements.*
>
> We understand that this comment is in regards to the contents of Section 7. During the rebuttal period, we have performed new experiments with super-moves and dynamic action spaces, which show improvement over the baseline results reported in the manuscript.
>
> Our setup of dynamic action spaces looks as follows. We note that in many sequences of AC paths, a common strategy that the agent employs is to select several conjugation moves (AC$'$2) in a row followed by a single concatenation move (AC$'$1). We increased the size of the action space to include actions that perform conjugations by arbitrary words of generators and their inverses, followed by a single concatenation. This significantly enhances the size of the action space from the original $12$ to several thousands. In the beginning, however, we keep all of these new actions masked, letting the agent play the game with the original AC moves.
>
> As the agent solves any new presentation, we scan through the path it found, look for the sub-sequences of moves of the type described above and unmask only the actions corresponding to these sub-sequences. We repeat this process after every 100 rollout phases of the PPO training loop. With this approach, the agent solves $13$ new presentations from the Miller--Schupp dataset that were previously unsolved by greedy search and our reinforcement learning agents with static action spaces.
>
> These results provide empirical evidence on the usefulness of dynamic action spaces and supermoves. We commit to adding these results to Section 7 of the camera-ready version of the paper, along with providing paths for the $13$ newly solved presentations. We will also include the JAX code used for these experiments along with our submission.

---

### Official Review · Reviewer_wqtT · 2025-07-04

**Clarity:** 2
**Significance:** 4
**Originality:** 3
**Rating:** 4
**Confidence:** 3

**Summary:**

The paper presents a reinforcement learning program to find solutions and counterexamples to several long-standing conjectures from combinatorial group theory. The paper goes into detail about the Andrews–Curtis trivialization problem (conjecture), framing it as a reinforcement learning task. The corresponding environment is characterized by long horizons, sparse rewards, and also an intrinsic distribution of ``problem hardness'', which is native to the framing of the conjecture. For a large class of examples, through intrinsic and path-based measures, the distribution of problem hardness is characterized and finally various RL agents are deployed on problems of varying hardness and the resulting observations are used to improve these agents to dynamically adapt to problem difficulty.

### Problem description
A group presentation, $\pi = \langle x\_1, x\_2, \dots, x\_n \mid r\_1, r\_2, \dots, r\_m \rangle$, consists of a set of generators $\\{ x\_i \\}\_{i=1}^n$ and a set of relators $\\{r\_j\\}\_{j=1}^m$, where each relator $r_j$ is a word in the alphabet $\\{x\_i^{\pm1} \\}\_{i=1}^n$. Consider the set of balanced presentations where $n = m$. The length of a presentation is the sum of the word lengths, $\sum\_i |r_i|$.

**AC conjecture.**
Every balanced presentation of the trivial group is *AC‐equivalent* to the trivial presentation, $\langle x\_1, \dots, x\_m \mid x\_1, \dots, x\_m \rangle$. That is, starting from any balanced presentation of the trivial group, one can obtain the trivial presentation by a finite sequence of the following *AC‐moves*:
- $r\_i \;\mapsto\; r\_i\,r\_j$, for $i\neq j$ (Nielsen–type move),
- $r\_i \;\mapsto\; r\_i^{-1}$,
- $r\_i \;\mapsto\; x\_j^{\pm1} r\_i x\_j^{\mp1}$.

The paper focuses on the case $m=2$.

With this presentation, ``checking'' whether a balanced presentation is AC-equivalent to the trivial presentation is by itself not an easy problem. This involves checking through all possible sequences of AC-moves as to whether the two presentations may be equivalent. This search problem can be formulated as checking whether two nodes are connected on a combinatorially large graph where two nodes are connected if they are equivalent up to a single AC move.

Problem difficulty is nicely captured by the length of the final path that connects a balanced presentation to the trivial presentation. Another measure of difficulty is the length of the longest presentation along the path that connects the candidate presentation to the trivial presentation.

The PPO agent trained by the paper aim to solve the connectivity problem and are trained on a dataset of ~1200 datapoints, with an accuracy of approximately 40%. However, since the RL environment is verifiable, i.e., a candidate path can be checked for whether it is correct or not, presumably this accuracy can be improved by considering even larger datasets.

The authors also present some cool theoretical results which are apparently guided by the empirical observations made in the experiments. However, the connection between these two was a little less clear to me.

**Questions:**

N/A

**Ethical Concerns:**

["NO or VERY MINOR ethics concerns only"]

**Limitations:**

Yes

**Quality:**

3

**Strengths And Weaknesses:**

The paper presents a very interesting use of RL for solving search problems on graphs in the context of the AC conjecture. The paper uses relatively simple algorithm (A2C / PPO) on relatively small sized datasets and already shows promising performance.

I think the writing of the paper can be improved a little bit. It isn't clear until the results section of the paper, which appears much later than I would have liked, the actual contributions of the paper. Some details are spelled out in the introduction, but not nearly enough to get a sense of what is going on until the results section. I like the description of the main results on page 12 of the appendix - the two theorems stated here need to be made just as clear in the main paper, and the implications of these results to those not familiar with the AC conjecture need to be spelled out more clearly. I would invest more effort into stating the problem very clearly once the AC conjecture and other abstractions are spelled out - that of certifying whether plausible counterexamples to the AC conjecture are indeed counterexamples, or whether they are trivializable. These are small things which may not need to be stated explicitly for someone familiar with the problem, but would really help someone from the broader audience understand the paper.

From a technical point of view, I think the paper is quite nice. I have three constructive criticisms:

1. I did not exactly understand how the empirical results in the paper laid the foundations for the theorems in the paper to be proved. It would be helpful to spell this out for the sake of bridging the two ``halves'' of the paper in a better fashion.

2. I really think the results in the paper would be supremely interesting if scaled up further. Perhaps there are computational bottlenecks that I don't understand as clearly, but running PPO / A2C on a 1e3 sized dataset of horizon length 200 seems like a fairly small setup. It should be clarified to a reader, or to someone intending to build upon these results what is the bottleneck in the current approaches: is it that it is hard to construct more synthetic examples, or is it an issue with scaling RL, or is it compute?

3. Furthermore, I did not fully understand why an *actual* search algorithm like MCTS or A* was not a part of the family of algorithms being trialed. These algorithms are really good at solving such search problems (as evidenced by a number of examples in the literature) and also they have the intrinsic ability of being scalable with more access to compute at test-time (i.e., just run longer depth searches).

I like the problem that was tackled in this paper, I think there are some suggestions above the authors can work on / clarifications to be made in the rebuttal that I would be open to hearing and revising my score on the basis of.

---

> ### Author Rebuttal · Authors · 2025-07-31
>
> Dear reviewer,
>
> Thank you for your thoughtful and constructive review. We appreciate the time you took to engage with our work and are pleased you liked the problem and overall direction of the paper. Moreover, you raise some excellent concerns, fixes for which we would be happy to edit into the final camera-ready version.
>
>
> 1. First, we would be happy to add the following new paragraph at the end of Section 3.1 to clarify the problem statement for a broader audience as you suggested.
>
> *The conjecture has been open for several decades. Many potential counterexamples of the conjecture have been proposed in the literature, often in the form of infinite families of presentations of the trivial group. In order to study whether a presentation is consistent with the conjecture, one aims to find its trivialization. Previous studies have used breadth-first search and genetic algorithms to find trivializations for some presentations, while the status of many presentations still remains open. In this work, we explore the question of finding trivializations for these open cases --- the so-called potential counterexamples, using reinforcement learning techniques.*
>
>
> 2. Next, we turn our attention to your following comment.
>
> *I did not exactly understand how the empirical results in the paper laid the foundations for the theorems in the paper to be proved. It would be helpful to spell this out for the sake of bridging the two `halves' of the paper in a better fashion.*
>
> We have edited Section 5.3 and Appendix A to convey this connection better. Indeed, the empirical results in the paper were crucial to the discovery of theorems in the more mathematical part of the paper. When our agents discovered any paths for specific presentations of Miller--Schupp or Akbulut-Kirby series, we first re-analyzed their solution paths of elementary AC-moves, in terms of substitution moves. Then we identified any patterns and generalized them to find new results for infinite families.
>
> Applying this procedure to our agents' solution paths for $MS(1,w)$ with $\text{length}(w) \leq 7$ led to the general result of Theorem 5.1. Similarly, analyzing the path connecting $\text{AK}(5)$ to a length-16 presentation led to length reduction of $\text{AK}(n)$ for all $n \geq 5$.
>
> We have added the following paragraph to Section 5.3.
>
> *The mathematical contributions of our agents fall into two categories. First, the agents discovered explicit paths that trivialized hundreds of previously-open presentations in the Miller--Schupp series. Second, we analyzed some paths of AC moves in terms of substitution moves (defined in Appendix A.1) to discover new patterns, which we then generalized to find results for infinite families. We emphasize that these results would not have been possible without the empirical results reported in this work. Here, we present a summary of our results, with a more detailed example of this process given in Appendix A.*
>
> In Appendix A, we have added a more detailed example of our workflow showing how it was used to obtain the result regarding length reduction of $AK(n)$. We have also included how the intermediate step of interpreting our agents' paths in terms of substitution moves is crucial as identifying patterns in the raw sequences of AC paths discovered by the agent is certainly more difficult. Hence, our appendix reflects a good example of how AI agents and human mathematicians can work together towards the goal of scientific discovery.
>
>
> 3. Now we discuss the issue related to scaling data, horizon length, and compute. We turn our attention first to the issue of dataset size.
>
> - The natural approach to creating a large dataset would be to sample random balanced presentations $\langle x_1, x_2 \mid r_1, r_2 \rangle$ and check if the given presentation is a presentation of the trivial group. There are two obstacles in following approach. First, as the length of the presentation increases, the percentage of presentations that present the trivial group drops substantially. The second and more problematic issue is that, given a random presentation, it is in general not possible to determine if it is a presentation of the trivial group. (Practically, this means that while there are algorithms to determine this, they are inefficient and do not terminate on some inputs.)
>
> - Another natural approach would have been to include more examples from the Miller--Schupp with larger $n$ and $w$. Amongst Miller--Schupp presentations, we notice a trend in Figure 2b. All algorithms start to struggle as the value of n is increased from 1 to 7. (A similar pattern also exists as a function of length($w$). The current manuscript does not have this plot, but we will be happy to include it, if needed.) We have noted in our experiments that this trend continues as the values of $n$ and $\text{length}(w)$ are increased further. Hence most of the presentations of Miller--Schupp series with larger values of n and $\text{length}(w)$ will require extremely long paths and so, given our agent was already struggling with easier presentations, adding more difficult presentations seemed unlikely to help.
>
> - A third natural strategy to increase the dataset size could be to start from a presentation of the trivial group, and apply a random sequences of AC moves to obtain a large dataset of presentations. In our observation, this technique ends up mostly producing uninteresting presentations. They are either identical to the initial representation except that the relations have been rotated and/or inverted or extremely long with an easy-to-discover path back to the starting presentation. We also expect that these presentations will be encountered by an RL agent in its rollouts in any case, and hence, adding them provides no significant advantages.
>
> - We also remark that in the case of Andrews--Curtis environment, the hardest cases are millions of times harder than generic cases, as shown by Bridson and Lishak in the case of 3 and higher number of generators. We have recently made progress in creating a 2-generator analogue of their families of presentations. This could serve as an excellent benchmark for the development of novel reinforcement learning algorithms in the future. In the current paper, however, we avoided training datasets of easy presentations, (such as those that are close to the goal state), as gaining high levels of performance on such a dataset could be misleading.
>
> We appreciate your feedback that we should communicate the issues around dataset creation in the paper. We will be happy to include a paragraph in the Limitations section, with more detailed accounting in an appendix.
>
> We now turn our attention to the issue of small value of horizon length. We experimented with a wide range of horizon length values, observing the following scaling trend: at a fixed level of performance in terms of the number of presentations solved, the number of environment interactions required to train an agent scales linearly with horizon length. The slope of the trend is positive and depends on the level of performance. This indicates that as we increase the horizon length, we need larger amounts of compute time to solve the same number of presentations. As we are a small academic lab, we circumvented the requirements of larger compute for training at longer horizon length by varying the horizon length during training. As reported in the paper, this modification helped us find some new trivializations.
>
> Finally, we trained models of different sizes for varying amounts of number of environment interactions. The largest model whose result is reported in Section 5.2 is a six layer residual network actor-critic agent, trained for 1 billion environment interactions. If recommended, we will be able to train much larger models for much longer amounts of time. However, we suspect that the main limitations in the agents' performance comes from the current algorithms' struggles with sparse-rewards, long-horizon environments. For these reasons, we turned our focus to developing new algorithms involving supermoves and dynamic action spaces, which we describe in Section 7.
>
> We understand that we did not discuss these issues satisfactorily in the submitted draft. As the camera-ready version allows for one more page, we will be able to incorporate your feedback quite comfortably in the paper.
>
>
> 4. Regarding the testing of other search algorithms, we did some experimentation involving other algorithms but found them universally less successful than greedy search.
>
> Looking at $A^*$ search as a specific example, this is essentially a half-way option between breadth first and greedy search, where the priority queue is sorted by the sum of the path length $p$ and presentation length $l$. We tried a search algorithm that sorted the priority queue using $r * p + l$ for a variety of choices of $r \geq 0$ and tested how many MS presentations the search algorithm solved. What we found was that, in all cases, decreasing $r$ improved the number of presentations solved and so we ended up just setting $r = 0$ at which point we end up back with greedy search.
>
> As for MCTS, we observed that MCTS spends a meaningful amount of time exploring different paths which do not lead to a trivialization solution. In contrast, the length heuristic function of greedy search helps find AC trivializations much faster, and with a much smaller usage of system memory.
>
> We will be happy to incorporate comments on these findings in the camera-ready version of the draft, providing more clarity for the preference of breadth-first search (which finds the shortest paths) and greedy search (which solves the most presentations) in our plots showing comparison with reinforcement learning techniques.

---

### Decision · Program_Chairs · 2025-09-17

**Decision:**

Accept (poster)

**Comment:**

This paper presents an ambitious case study on applying reinforcement learning (RL) techniques to a long-standing mathematical problem—the Andrews–Curtis conjecture. It frames this as a sparse-reward, long-horizon RL environment and analyzes the difficulty of solving instances using PPO and A2C agents. The authors contribute novel mathematical results, including resolving previously open instances, and propose techniques such as supermoves and dynamic action spaces to address problem hardness.

This was a borderline case. Initially, reviewers leaned toward rejection, noting that while the setting is intriguing, the RL contributions felt limited and the focus was more on mathematical outcomes. However, the authors engaged seriously during the rebuttal, providing additional experiments, comparisons to other baselines (e.g., DQN, AlphaZero), and empirical support for their proposed techniques. These additions shifted reviewer opinion, and several now support acceptance.

We recommend acceptance, trusting in good faith that the authors will revise the paper accordingly. If the review process allowed for conditional acceptance, we would have required these revisions as part of the final decision.